# Reasoning with Latent Thoughts: On the Power of Looped Transformers

**Nikunj Saunshi**[1]**, Nishanth Dikkala**[1]**, Zhiyuan Li**[1,2]**, Sanjiv Kumar**[1]**, Sashank J. Reddi**[1]
{nsaunshi, nishanthd, lizhiyuan, sanjivk, sashank}@google.com
[1]Google Research, [2]Toyota Technological Institute at Chicago

## Abstract

Large language models have shown remarkable reasoning abilities and scaling laws suggest that large parameter count, especially along the depth axis, is the primary driver. In this work, we make a stronger claim — many reasoning problems require a large depth but not necessarily many parameters. This unlocks a novel application of *looped models for reasoning*. Firstly, we show that for many synthetic reasoning problems like addition, $p$-hop induction, and math problems, a $k$-layer transformer looped $L$ times nearly matches the performance of a $kL$-layer non-looped model, and is significantly better than a $k$-layer model. This is further corroborated by theoretical results showing that many such reasoning problems can be solved via iterative algorithms, and thus, can be solved effectively using looped models with nearly optimal depth. Perhaps surprisingly, these benefits also translate to practical settings of language modeling — on many downstream reasoning tasks, a language model with $k$-layers looped $L$ times can be competitive to, if not better than, a $kL$-layer language model. In fact, our empirical analysis reveals an intriguing phenomenon: looped and non-looped models exhibit scaling behavior that depends on their effective depth, akin to the inference-time scaling of chain-of-thought (CoT) reasoning. We further elucidate the connection to CoT reasoning by proving that looped models implicitly generate *latent thoughts* and can simulate $T$ steps of CoT with $T$ loops. Inspired by these findings, we also present an interesting dichotomy between reasoning and memorization, and design a looping-based regularization that is effective on both fronts.

## 1 Introduction

Language models have shown a lot of promise in solving problems that require strong reasoning abilities like math, coding, common sense reasoning and logical puzzles (Brown et al., 2020; Team et al., 2023). This has sparked interest in developing techniques to improve reasoning on harder problems (Wei et al., 2022b) and has inspired theoretical studies on how Transformers are able to perform reasoning (Feng et al., 2024; Sanford et al., 2024a). Reasoning abilities are often emergent in larger language models (Wei et al., 2022a) – this aligns with various scaling laws (Kaplan et al., 2020; Hoffmann et al., 2022; Allen-Zhu & Li, 2024) that show that the performance of language models is very strongly dependent on the model size (i.e., number of parameters) and much lesser on other architectural design choices. However, recent works have started to question this view. Ye et al. (2024) argue that scaling laws for reasoning are more subtle, and *depth is very important* in addition to parameter count – at the same parameter count, deeper but shallower models are better. This is a deviation from the conventional scaling law wisdom, but it intuitively makes sense because reasoning problems often requires multi-step compositional thinking, and thus, depth can play a crucial role.

In this work, we make a stronger claim – while depth is important, many reasoning problems do not necessarily require a lot of parameters. How does one solve reasoning problems with large depth but few parameters? We argue that *looped models* are perfectly suited, where a function, parameterized with few parameters, is iteratively applied on the input. This leads us to our first important claim:

*Claim 1: Many reasoning problems require depth but not necessarily parameters. That is, they can be solved via looped models*

Looped models have been studied in the literature for parameter efficiency (Lan et al., 2020), adaptive compute (Dehghani et al., 2018), equilibrium models (Bai et al., 2019) and for in-context learning (Yang et al., 2023; Gatmiry et al., 2024a). In this work, we **initiate the study of looped models in the context of reasoning**. Admittedly, reasoning is not very well-defined and can be of various forms (Sun et al., 2023). Acknowledging this hurdle, we focus on a non-exhaustive list of problems that intuitively require reasoning and that are inspired by reasoning benchmarks. Throughout the paper, we use the notation $(k \otimes L)$ to denote a $k$-layer model looped $L$ times (precise definition in Section 2), which has the same number of parameters as a $(k \otimes 1)$ model and same flops as a $(kL \otimes 1)$ non-looped model (see Figure 5). As a first step towards connecting looped models and reasoning, we empirically evaluate looped models on several simple reasoning tasks in the literature (e.g. Section 2). Perhaps surprisingly, we find that a $(k \otimes L)$ looped models does almost as well as, if not better than, a non-looped model $(kL \otimes 1)$ that has the same effective depth but $L$ times more parameters on these reasoning tasks. The looped model is also significantly better than a $(k \otimes 1)$ model which has the same number of parameters. Our theoretical results on the expressiveness of looped models in representing iterative algorithms with short description further corroborate these empirical findings and provide strong support for our claim. This naturally raises an important question: do looped models benefit language modeling in a similar manner?

> *Claim 2: For language modeling, looped models have an inductive bias towards good reasoning despite having worse perplexity and memorization to an iso-flop non-looped model*

For the above claim, we again train a $(k \otimes L)$ looped model on causal language modeling and compare it to the iso-param $(k \otimes 1)$ and iso-flop $(kL \otimes 1)$ non-looped baselines. While the looped model improves over the iso-param baseline, perhaps unsurprisingly, it ends up with worse perplexity than iso-flop baseline, since perplexity depends strongly on number of parameters. However, the downstream evaluations reveal an intriguing trend: looped models have a tendency to improve tasks that require reasoning a lot more than memorization tasks. Specifically, the looped model has reasoning performance much closer to the iso-flop baseline, sometimes even exceeding it despite having $L$ times fewer parameters and worse perplexity. This contrasting behavior between the pretraining and downstream metrics has been a subject of study lately (Saunshi et al., 2022; Liu et al., 2023) and is attributed to the *inductive biases* introduced due to different architectures and training algorithms. Our empirical analysis also uncovers an interesting phenomenon: accuracy on downstream tasks scale as logarithm of the effective depth. In particular, more loops enhances performance, and the relative benefit of loops is higher for tasks that require more reasoning. This is conceptually similar to inference time scaling discovered for CoT, but with looping as a central component. To further elucidate this interesting relationship of looped models with CoT, we present the following claim.

> *Claim 3: Looped models generate latent thoughts and can, in theory, simulate CoT reasoning*

Note that CoT reasoning gives the model more time and compute by generating multiple thought tokens before the answer, and it has powered the recent paradigm of inference-time scaling for "thinking" models like O1 and DeepSeek's R1 (Guo et al., 2025). We make an observation about CoT reasoning – it is essentially a looped model that generates 1 thought token in each iteration. However, looped models seem to be much more powerful, since they can generate multiple *latent thoughts* in each iteration. We translate this intuition into a theoretical result about how looped models can simulate CoT reasoning. Motivated by these findings, we propose a regularization scheme that aims to tap into the inductive bias of looped models towards reasoning.

*Claim 4: Looping-inspired regularization can leverage this inductive bias towards better reasoning*

With the backdrop of these claims, we concretely present the contributions of the paper below:

- In this paper we study looped models – multilayer models with weight sharing – and their role in reasoning. In particular, we compare a $k$-layer model looped $L$ times, denoted by $(k \otimes L)$, with an *iso-param* $(k \otimes 1)$ non-looped model with $k$ layers and an *iso-flop* $(kL \otimes 1)$ model with $kL$ layers and $L$ times more parameters.
- We conduct experiments on synthetic reasoning tasks like addition, $p$-hop induction and GSM-style math word problems in Section 2. For these tasks, we surprisingly find that iso-flop looped models, despite having way fewer parameters, can nearly match or outperform a non-looped

model. Supporting these experiments, in Section 5 we present theoretical results for why looped models can solve such problems with almost optimal depth.

- In Section 3, we train looped models on causal language modeling at 1B parameter scale. Here, we show that looped models have an inductive bias towards doing well on reasoning benchmarks, despite having much worse perplexity. This finding is novel, since most prior work on looping focused more on perplexity metrics rather than downstream reasoning tasks. We validate this inductive bias by visualizing the perplexity vs downstream performance plots as training process. Additionally, we show that looped models demonstrate good scaling behavior on various benchmarks as the number of loops are increased, akin to CoT reasoning. Finally, we show that looped models, along with a scratchpad, can simulate chain-of-thought reasoning.
- Inspired by this inductive bias, in Section 4, we propose a regularization that encourages layers to be more similar to each other. We find that training with such a regularization inherits the inductive bias of looped models towards reasoning without affecting perplexity.

## 2 LOOPED MODELS ON SIMPLE REASONING TASKS

We first explore our hypothesis of looped models helping reasoning tasks on a set of tasks constructed in a procedural manner. The illustrative reasoning tasks we consider are: $n$-ary addition, $p$-hop induction head that tests the model's ability to track back for $p$ steps, and i-GSM which consists of synthetically constructed grade-school math problems. While these obviously do not cover the whole spectrum of reasoning problems, they provide useful insights into looped models and provide a basis for the theoretical results in Section 5.

**Looped models.** While many variants of looped model have been proposed (Lan et al., 2020; Dehghani et al., 2018; Giannou et al., 2023; Yang et al., 2023; Mohtashami et al., 2023), we use the vanilla version for simplicity. For any sequence-to-sequence function $f$, we denote $f^{(L)} = f \circ f \cdots \circ f$ to be the function that is $f$ looped $L$ times. In general, the looping mechanism is independent of architectural choices for $f$. For the rest of the paper, we typically use $f$ to denote a $k$-layer Transformer backbone of a model. Thus, $f$ looped $L$ times is the same as a $kL$ layer model with weight sharing between all $L$ blocks of $k$ consecutive layers. We denote such looped models with the notation $(k \otimes L)$. Please refer to Figure 5 for a succinct illustration of the looping mechanism. Section 5.1 provides a more formal definition of looped transformers that is used for theoretical analysis.

### 2.1 EXPERIMENTS WITH SIMPLE REASONING PROBLEMS

$n$**-ary addition.** We consider the problem of adding $n$ numbers with 3 digits each. Addition is popular in the literature to study aspects of reasoning with Transformers, such as use of scratchpad (Nye et al., 2021), chain of thought reasoning (Lee et al., 2024; Li et al., 2024) and length generalization (Cho et al., 2024). One reason for its popularity is that addition can have algorithmic solutions, which is a feature of many reasoning problems. For our experiments, we train on a uniform mixture on numbers of operands $n \in \{2, 4, 8, 16, 32\}$ and sample each 3-digit operand uniformly at random between $[0, 999]$. We train all models directly on the input-output pair, without any chain-of-thought steps. For instance, for $n = 4$, we have Input: "$315 + 120 + 045 + 824 =$" ; Output = "$1304$".

We train a standard Transformer-based baseline $(12 \otimes 1)$ model with 12 layers. Please refer to Appendix B.1 for details on the training setup. We also train $(k \otimes 12/k)$ looped model and an iso-param $(k \otimes 1)$ baseline models for comparison, and vary $k \in \{2, 3, 4, 6\}$. All trained models are finally evaluated separately on each of $n \in \{8, 16, 24, 32\}$ to measure accuracy on increasingly difficult problems. Results are presented in Table 1. We find that, while the shallower baselines $(k \otimes 1)$ degrade with lower $k$, the looped model $(k \otimes 12/k)$ performs very well, and nearly matches the iso-flop $(12 \otimes 1)$ baseline. In fact, even a 1-layer network looped 12 times is able to solve this, despite using merely $1/12^{th}$ of the parameters of the baseline. This suggests the addition problem primarily requires depth, but not necessarily more parameters.

$p$**-hop induction.** The $p$-hop problem is a synthetic induction task studied in Sanford et al. (2024b), who were inspired by the analysis of induction heads from Elhage et al. (2021). Specifically, given a sequence of letters $\boldsymbol{v} = (v_1 \ldots v_n)$ from an alphabet $\Sigma$, an induction head tries to find the penultimate occurrence of $v_n$ (from left to right) in the sequence and output the character

Table 1: Accuracy of looped and non-looped models on the addition problem (left) and $p$-hop induction (middle), and i-GSM (right) as described in Section 2. **Left.** For addition, we report accuracies for different number of operands ($n$). For all budgets, a $(k \otimes 12/k)$ looped model is significantly better than the iso-param $(k \otimes 1)$ model and also nearly as good as the non-looped iso-flop $(12 \otimes 1)$ baseline model. **Middle.** The findings are very similar for the $p$-hop problem for different values of $p$. Note that a random guessing baseline would get at least 25% accuracy (since only 4 choices for answer). This suggests that depth via looping and small number of parameters is very effective for these problems. **Right.** Accuracy on i-GSM task from Section 2. $(k \otimes L)$ looped model is significantly better than the iso-param $(k \otimes 1)$ model and performs as well as non-looped iso-flop $(kL \otimes 1)$ model.

**Addition of $n$ numbers**

| | Params / FLOPs | $n = 24$ | $n = 32$ |
|---|---|---|---|
| Base ($12 \otimes 1$) | 12x / 12x | **100.0** | **100.0** |
| *1 layer model* | | | |
| Base ($1 \otimes 1$) | 1x / 1x | 0.1 | 0.0 |
| Loop ($1 \otimes 12$) | 1x / 12x | **99.9** | **99.6** |
| *2 layer model* | | | |
| Base ($2 \otimes 1$) | 2x / 2x | 49.3 | 38.8 |
| Loop ($2 \otimes 6$) | 2x / 12x | **99.7** | **99.5** |
| *3 layer model* | | | |
| Base ($3 \otimes 1$) | 3x / 3x | 69.2 | 60.7 |
| Loop ($3 \otimes 4$) | 3x / 12x | **97.0** | **96.6** |

**$p$-hop with $n$ tokens**

| | Params / FLOPs | $p = 16$, $n = 256$ | $p = 32$, $n = 256$ |
|---|---|---|---|
| Base ($6 \otimes 1$) | 6x / 6x | **99.9** | **99.6** |
| *1 layer model* | | | |
| Base ($1 \otimes 1$) | 1x / 1x | 48.9 | 49.0 |
| Loop ($1 \otimes 6$) | 1x / 6x | **99.9** | **99.5** |
| *2 layer model* | | | |
| Base ($2 \otimes 1$) | 2x / 2x | 68.8 | 59.4 |
| Loop ($2 \otimes 3$) | 2x / 6x | **99.9** | **99.8** |
| *3 layer model* | | | |
| Base ($3 \otimes 1$) | 3x / 3x | 97.2 | 73.0 |
| Loop ($3 \otimes 2$) | 3x / 6x | **99.9** | **99.5** |

**i-GSM**

| | Params / FLOPs | Accuracy |
|---|---|---|
| Base ($8 \otimes 1$) | 8x / 8x | **73.2** |
| *1 layer model* | | |
| Base ($1 \otimes 1$) | 1x / 1x | 24.5 |
| Loop ($1 \otimes 2$) | 1x / 2x | 52.3 |
| Loop ($1 \otimes 4$) | 1x / 4x | 69.9 |
| Loop ($1 \otimes 8$) | 1x / 8x | **73.2** |
| *2 layer model* | | |
| Base ($2 \otimes 1$) | 2x / 2x | 54.0 |
| Loop ($2 \otimes 2$) | 2x / 4x | 66.9 |
| Loop ($2 \otimes 4$) | 2x / 8x | **73.6** |
| *4 layer model* | | |
| Base ($4 \otimes 1$) | 4x / 4x | 71.3 |
| Loop ($4 \otimes 2$) | 4x / 8x | **71.6** |

immediately succeeding it. The $p$-hop problem generalizes this idea to sequentially hop $p$ times. Intuitively, the $p$-hop problem tests a model's ability to recursively backtrack and retrieve the answer for a given query. This is reminiscent of the reasoning abilities required to solve reading comprehension kind of problems. We present the formal definition of the $p$-hop problem in Definition B.1. We perform experiments with looped and non-looped models on the $p$-hop problem, with alphabet size set to 4 and sequences of length 256, and vary $p$ between 16 and 32 to control the problem difficulty. Our observations are presented in Table 1. Similarly to our findings on the addition task, reasonably deep looped models perform as well as the baseline using much fewer parameters.

**i-GSM (Synthetic Grade School Math Problems).** Inspired by Ye et al. (2024), we built our own version of grade-school level math word problems. While we follow many of the design guidelines of Ye et al. (2024), we make a few simplifying changes. We generate the math problem as a DAG of arithmetic computations modulo 7, and restrict the depth of the graph to 4. For simplicity, we retain the problems in the symbolic form and do not map them to English (e.g., see Table 8) [1] We train models of depths 1, 2, 4 and 8 and compare them with different looped variants in Table 1. The answer is computed modulo 7. Hence, a random guessing baseline would get at least 14% accuracy. Appendix B.1 has more details. Remarkably, we again observe that a depth $k$ model looped $L$ times matches or outperforms a depth $kL$ model and far outperforms a depth $k$ non-looped model. This suggests that even a more complex and realistic looking reasoning problem does not need too many parameters and can benefit from depth via looping.

# 3 LANGUAGE MODELING WITH LOOPED MODELS

In this section, we pretrain and evaluate looped models for causal language models. We train models on 250B tokens of the Pile dataset (Gao et al., 2020) and use a 24-layer 1B parameter model for most experiments, motivated by the setting in Tay et al. (2022) (refer to Appendix B.2 for more details).

## 3.1 EXPERIMENTS WITH LANGUAGE MODELING

For causal language modeling, we pretrain various looped models on the standard GPT2-style next token prediction objective (Radford et al., 2019). We train models with different parameter budgets to make sure that the findings are robust. We remind the reader that the notation $(k \otimes L)$ corresponds to a $k$ layer model looped $L$ times. For each setting, we compare 3 models: **(a)** $(24 \otimes 1)$: 24-layer 1B model, **(b)** $(k \otimes 1)$: $k$-layer model with the same configuration as the 24-layer model for other dimensions, **(c)** $(k \otimes 24/k)$: $k$-layer model looped $24/k$ times to match the parameter count of (b)

---

[1]Similar to Ye et al. (2024), the simplified setting still allows for over 7 billion unique solution templates.

Table 2: Downstream evaluations for language models trained on the Pile dataset. Comparisons include a 24-layer 1B-parameter baseline model, iso-flop looped models ($k \otimes 24/k$) for various parameter budgets $k$, and the corresponding iso-param baselines ($k \otimes 1$). Downstream evaluations are averaged over tasks within 4 task groups. We also include the % Gap metric for each $k$ to measure the gap between the iso-param and iso-flop baselines that is covered by the looped model (see Equation (1)). Overall the looped models are worse on perplexity and closed book QA (memorization benchmarks), but the % Gap is much higher for task groups that require reasoning (open book QA, math word problems). In fact for reasoning primitives, which are purely testing for reasoning skills, the looped models are much better than the 1B baseline for all $k$, despite having $24/k \times$ fewer parameters.

| | Params / FLOPs | Perplexity (↓) (validation) | Closed Book QA (↑) (4 tasks) | Open Book QA (↑) (5 tasks) | Math Word Problems (↑) (6 tasks) | All Tasks Average (↑) (15 tasks) | Reasoning Primitives (↑) (4 tasks) |
|---|---|---|---|---|---|---|---|
| **24 layers** | | | | | | | |
| Baseline | 24x / 24x | 7.40 | 11.2 | 33.9 | 29.3 | 26.0 | 47.5 |
| **12 layers** | | | | | | | |
| Base ($12 \otimes 1$) | 12x / 12x | 8.16 | 8.2 | 26.9 | 26.7 | 21.8 | 35.7 |
| Loop ($12 \otimes 2$) | 12x / 24x | 7.90 | 9.3 | 30.8 | 34.3 | 26.5 | 51.2 |
| % Gap | | 34 % | 37 % | 56 % | 282 % | 110 % | 131 % |
| Middle Loop ($4 \otimes 1, 4, 1$) | 12x / 24x | 7.81 | 11.0 | 32.3 | 28.3 | 25.0 | 56.5 |
| % Gap | | 46 % | 94 % | 78 % | 62 % | 95 % | 176 % |
| **8 layers** | | | | | | | |
| Base ($8 \otimes 1$) | 8x / 8x | 8.75 | 6.3 | 22.7 | 17.1 | 16.1 | 33.0 |
| Loop ($8 \otimes 3$) | 8x / 24x | 8.19 | 8.5 | 30.8 | 28.4 | 23.9 | 55.3 |
| % Gap | | 41 % | 44 % | 72 % | 92 % | 78 % | 153 % |
| **6 layers** | | | | | | | |
| Base ($6 \otimes 1$) | 6x / 6x | 9.25 | 4.0 | 19.3 | 17.7 | 14.6 | 24.1 |
| Loop ($6 \otimes 4$) | 6x / 24x | 8.42 | 8.2 | 28.7 | 29.8 | 23.7 | 56.1 |
| % Gap | | 44 % | 58 % | 64 % | 104 % | 80 % | 136 % |
| **4 layers** | | | | | | | |
| Base ($4 \otimes 1$) | 4x / 4x | 10.12 | 1.8 | 13.8 | 9.7 | 9.0 | 19.4 |
| Loop ($4 \otimes 6$) | 4x / 24x | 8.79 | 6.7 | 26.2 | 24.8 | 20.4 | 56.9 |
| % Gap | | 48 % | 52 % | 61 % | 77 % | 67 % | 133 % |

and match the effective depth/FLOPs of (a). We run experiments for $k \in \{4, 6, 8, 12\}$ to ensure that the findings are robust. After pretraining on Pile, we evaluate the models on validation perplexity and on downstream benchmarks using $k$-shot evaluations. Results are summarized in Table 2

**Evaluation metrics.** We evaluate the models on perplexity metric after training is completed. Since there is growing evidence that perplexity, although very useful for training, is a narrow measure of model quality, we also track more holistic downstream evaluations (Liang et al., 2023). Thus, we evaluate the model on 4 important slices: closed book QA, open book QA, math word problems and reasoning primitives. These comprise of **19 different tasks** in total. We defer details for the evaluation benchmarks in Appendix B.3

For each task group $G$ from above, in Table 2 we report the average accuracy for that task group, denoted by $\text{Avg}_G$. Furthermore, for each layer budget $k$, we report the **% gap** between the iso-param and iso-flop models that is covered by the looped model. More specifically

$$\% \text{ Gap} = \frac{\text{Avg}_G(k \otimes 24/k) - \text{Avg}_G(k \otimes 1)}{\text{Avg}_G(24 \otimes 1) - \text{Avg}_G(k \otimes 1)}. \tag{1}$$

This measures how effectively looping can bridge the gap between iso-param and iso-flops baselines. Implicitly, it measures how different task groups behave when a model only has a few parameters but is given depth through looping. % Gap being closer to 0% means that providing depth via looping does not benefit the task group, and number of parameters is the most important factor. On the other hand, % Gap closer to 100% means parameter count matters much less for the task group, and that depth via looping is more essential.

**Perplexity results.** Firstly we notice that all $(k \otimes 24/k)$ looped models have better perplexity compared to the iso-param $(k \otimes 1)$ baseline, but worse perplexity compared to the non-looped 24-layer baseline. The looped models only covers up roughly $34 - 50\%$ of the perplexity gap between the iso-param and iso-flop baselines for various values of $k$. This perplexity gap is not too surprising since the looped model has $24/k$ times fewer parameters, and thus, lower capacity than the 24-layer baseline. This was also been observed in prior works (Lan et al., 2020; Mohtashami et al., 2023)

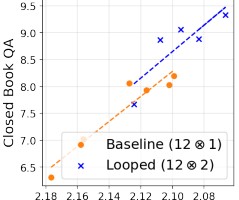 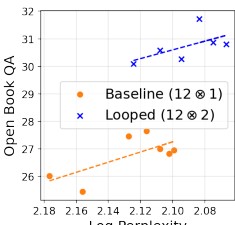 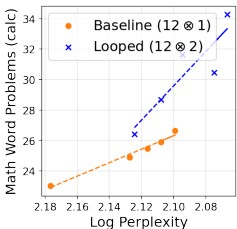 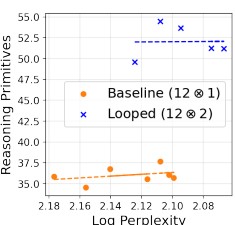

Figure 1: Downstream evaluation for various task groups on the x-axis, vs validation log perplexity on the y-axis (reversed), as training proceeds. We compare a 12-layer baseline model $(12 \otimes 1)$ with a looped model $(12 \otimes 2)$. For closed book QA (memorization) tasks looping has very similar trends to baseline. For open book QA tasks and math word problems, looping has much better downstream performance at an equivalent log perplexity.

and is the primary reason looped models have been ignored for language modeling. However, as we shall see shortly, the downstream metrics paint a more interesting and favorable picture.

**Results on QA tasks.** We first consider closed book and open book QA categories in Table 2. Closed book QA tasks are primarily testing the model's memorization abilities. Open book QA on the other hand tests the model's ability to infer the answer from the additional context that is provided. Thus, intuitively, open book QA tasks require more reasoning. Firstly we notice that the % Gap for closed book QA (memorization) is very similar to % Gap for perplexity. The % Gap for open book QA, on the other hand, is much higher for all parameter budgets. This suggests that looped models relatively improve contextual QA much more than memorization based QA.

**Math problems and reasoning primitves.** We also present the % Gap for the math word problem in Table 2. Surprisingly, we find that $(k \otimes 24/k)$ looped model can almost match the baseline $(24 \otimes 1)$ model for $k \geq 6$, despite having $k$ times fewer parameters. In fact, the 12 layer model looped twice is even signficantly better $(34.3)$ than the 24 layer baseline $(29.3)$, despite having 50% of parameters and worse perplexity; suggesting that looping disproportionately improves mathematical reasoning.

To better understand the effect on reasoning, we direct the readers attention to the evaluations for reasoning primitives in Table 2. The results are quite remarkable: $(k \otimes 24/k)$ **looped models are better than the iso-flop baseline** $(24 \otimes 1)$ **at reasoning primitives, for all values of** $k$. This is a priori very surprising, since these are synthetic generated tasks and have nothing to do with the pretraining data or the model architecture. Thus, solving these tasks necessarily requires reasoning from context, and memorization abilities will not help here. These results clearly suggest that looped models have a bias towards improving reasoning, despite having worse perplexity and memorization.

## 3.2 INDUCTIVE BIAS TOWARDS REASONING

In this section, we formalize the inductive bias by plotting the perplexity vs downstream metric iso-plots, as introduced in Saunshi et al. (2022). Section 3.1 showed that looped models have *higher than expected* performance on reasoning problems. However, since looped models are worse on perplexity, it is hard to make a direct comparison between various models. One way to bring parity between models is to look at their downstream performances at the same validation pretraining loss (Liu et al., 2023). Saunshi et al. (2024) proposed plotting pretraining loss vs downstream metrics as training proceeds, as a way to study the inductive bias of various methods. For each model, we evaluate the log perplexity and downstream metrics at every 20k steps, starting from 120k steps. We plot these values in a scatter plot and fit a linear function with log perplexity and the corresponding downstream metric being input and output respectively. Please refer to Figure 1 for two sets of isoplots. For all values of $k$, we observe the following:

- The isoplots for $(k \otimes L)$ looped model and $(k \otimes 1)$ baseline are very aligned for closed book QA tasks (if extrapolated). This suggests that log perplexity is a very strong indicator of downstream performance on memorization based tasks.
- For open book QA and math word problems, the isoplot line for the looped model is always higher than the baseline model. This suggests that at the same log perplexity, looped models will tend to have higher evaluation on these tasks that require more reasoning.

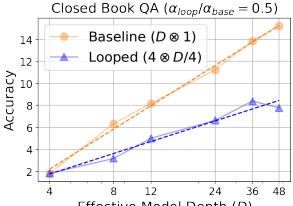 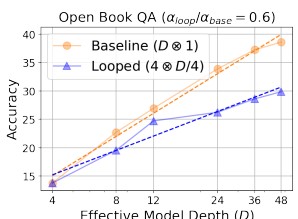 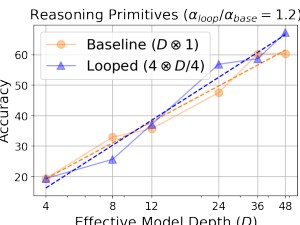

Figure 2: Scaling behavior for various task groups as the effective depth increases (also see Figure 7). The blue curve shows how performance scales for $(4 \otimes D/4)$ looped models as the number of loops increases, without increasing parameters. The orange curve visualizes the scaling behavior of $(D \otimes 1)$ which increases the depth by adding fresh parameters. For reasoning primitives, the looped model scales as well, or even better, than the baseline despite having $D/4$ fewer parameters.

- For reasoning primitives, there is a stark difference between looped and baseline models. The looped model seems to have good performance at most points in training.

This suggests a strong inductive bias of looped models towards improving reasoning. We discuss connections to gradual stacking (Appendix B.4) approaches that have a similar inductive bias.

### 3.3 SCALING BEHAVIOR OF LOOPING

In this section, we discuss an intriguing scaling behavior of looping, specifically the impact of the number of loops on various evaluation metrics. In particular, we are interested in scaling behavior of: (a) accuracy as a function of number of loops and (b) comparison of looped and non-looped baseline of the same effective depth. To this end, we pretrain various looped language models of the form $(4 \otimes L)$, i.e., 4-layer model looped $L$ times, for $L \in \{1, 2, 3, 6, 9, 12\}$. To enable iso-FLOPs comparison, we also train baseline models of the form $(4L \otimes 1)$, which has $L$ times more parameters. For each task group, we plot the average accuracy as a function of the *effective depth*, i.e. $D = 4L$. From the results presented in Figure 2, we observe the following.

1. In both cases, we find that the accuracies for all task groups continue to increase with more loops/depth, although, unsurprisingly, the returns are diminishing with depth for both looped and non-looped models. Interestingly, for both looped and non-looped models, we found that one can fit a simple scaling law of the following form:

$$\text{Acc} = \alpha \log(D) + \beta, \tag{2}$$

where $D$ is the effective depth and $\alpha$ measures the impact of depth on downstream performance.

2. Furthermore, for each task group, we compute $\alpha_{loop}/\alpha_{base}$ to assess the relative impact of "depth via looping" compared to "depth via additional parameters". We find that more loops continue to help, and the relative benefit of loops is higher for reasoning tasks like open book QA and math problems. Remarkably, the impact of loops is even higher (1.19x) than impact of depth for reasoning primitives, which further consolidates the benefit of looped models for reasoning.

**Latent thoughts and connections to CoT reasoning.** We end this discussion with an important question: *why should we expect this interesting scaling behavior of looped models?* We argue that looped models have a strong connection to chain-of-thought (CoT) reasoning. Such a connection is insightful because recent works on thinking have shown CoT can demonstrate inference-time scaling behavior, where the accuracy on reasoning datasets continues to improve as the length of the models chain of thought increases; i.e., generating more thoughts leads to better reasoning.

To establish this connection, we make a simple observation about CoT reasoning – it is essentially a looped model that generates a single thought token in each iteration. However, looped models can be more powerful since they can generate multiple "latent thoughts" in each iteration. This can be visualized in Figure 8. We further translate this intuition into a theoretical result (see Section 5.4) about how looped models can in fact simulate CoT reasoning. Given this connection to CoT reasoning, it is believable that looping can scale well for harder reasoning. We believe that leveraging looping explicitly for inference-time scaling is a very promising future direction.

Table 3: Results for the 24-layer 1B model with and without the regularization introduced in Section 4. We try various block sizes $k$ motivated by the looped model settings from Table 2. Overall, regularization helps retain the inductive bias towards reasoning, with notable improvements on math word problems and reasoning primitives, without almost neutral perplexity.

| | Perplexity ($\downarrow$) (validation) | Closed Book QA ($\uparrow$) (4 tasks) | Open Book QA ($\uparrow$) (5 tasks) | Math Word Problems ($\uparrow$) (6 tasks) | All Tasks Average ($\uparrow$) (15 tasks) | Reasoning Primitives ($\uparrow$) (4 tasks) |
|---|---|---|---|---|---|---|
| Baseline | 7.40 | 11.2 | 33.9 | 29.3 | 26.0 | 47.5 |
| Regularized ($k = 4, \lambda_{\mathrm{reg}} = 1$) | 7.41 | 11.2 | 34.8 | 31.6 | 27.2 | 42.5 |
| Regularized ($k = 4, \lambda_{\mathrm{reg}} = 10$) | 7.38 | 12.5 | 36.2 | 36.4 | 30.0 | 57.2 |
| Regularized ($k = 6, \lambda_{\mathrm{reg}} = 10$) | 7.40 | 12.0 | 35.8 | 31.0 | 27.5 | 55.8 |
| Regularized ($k = 8, \lambda_{\mathrm{reg}} = 10$) | 7.43 | 11.3 | 34.4 | 32.8 | 27.6 | 56.3 |
| Regularized ($k = 12, \lambda_{\mathrm{reg}} = 10$) | 7.51 | 10.1 | 34.1 | 32.3 | 27.0 | 50.7 |

## 4 LOOPING-INSPIRED REGULARIZATION

In the previous section, we observed the looped models can improve reasoning with worse perplexity. Can we leverage this observation to improve reasoning without affecting perplexity? Here, we propose a simple approach: regularize the weights of the model to encourage them to be close to a looped model. This could have two advantages, (a) the model still has free parameters to improve perplexity, (b) the closeness to looped model can inherit the desirable inductive bias. In particular, if an $L$-layer model is denoted as $f_0 \circ f_1 \ldots, \circ f_{L/k-1}$, where each $f_i$ is a block of $k$ layers, we add a regularization term that makes all the weights of the block $f_i$ close to $f_{i+1}$ in terms of cosine similarity. For a parameter group $G$ (e.g. first feed-forward layer, or query matrix in Transformer), we use $\theta_G^{(0)}, \theta_G^{(1)}, \ldots, \theta_G^{(L-1)}$ to denotes the weights in all layers. Then the regularization term is

$$\mathcal{R}_G(k) = \frac{1}{L-k} \sum_{i=0}^{\frac{L}{k}-2} \sum_{j=0}^{k-1} \mathrm{Cosine}\left(\theta_G^{(ik+j)}, \theta_G^{((i+1)k+j)}\right) \quad (3)$$

The final loss function is a sum of the standard cross-entropy loss and the regularization term averaged over all groups, multiplied by a scalar hyperparameter. Let $\mathcal{G}$ denote the set of all parameter groups; $\mathcal{G} = \{\mathrm{Attn\text{-}Q}, \mathrm{Attn\text{-}K}, \ldots, \mathrm{FFN\text{-}W2}\}$

$$\mathcal{L} = \mathcal{L}_{\mathrm{xent}} + \lambda_{\mathrm{reg}} |\mathcal{G}|^{-1} \sum_{G \in \mathcal{G}} \mathcal{R}_G(k) \quad (4)$$

In the above formulation, $\lambda_{\mathrm{reg}} = 0$ would recover standard training and $\lambda_{\mathrm{reg}} \to \infty$ would converge to a fully looped model. Intermediate values of $\lambda_{\mathrm{reg}}$ will lead to "approximately" looped models. For instance, to emulate the $(4 \otimes 6)$ looped model setting, we use pick $k = 4$, $L = 24$ and a large regularization strength like $\lambda_{\mathrm{reg}} = 10$. All other hyperparameters are kept the same as baseline training for a fair comparison. We tried options other than cosine similarity, like $\ell_2$ norm, to bring the weights closer but found that cosine was more robust and effective.

**Cosine similarities.** Firstly, we check if the regularization had the right effect by measuring the cosine similarities between the successive blocks of $k$ layers at the end of training. We, indeed, find that for all parameter groups, the cosine similarity around 0.98 or higher (see Figure 3).

**Inductive bias.** To confirm the inductive bias of the regularizer, we visualize the log perplexity vs downstream isoplots for $\lambda_{\mathrm{reg}} = 10$ and baseline models in Figure 6. While the plots are similar for closed book QA, a strong inductive bias shows up for open book QA and reasoning problems. Crucially, the regularized model does well on reasoning without hurting perplexity (see Table 3).

## 5 THEORETICAL ANALYSIS FOR LOOPED MODELS

In this section, we present theoretical results to understand the phenomenon from the previous sections – *why can looped model with few parameters match an iso-flops non-looped baseline on reasoning problems?* While a complete theory is challenging, since "reasoning" is a very broad concept, the goal is to provide some intuition and formalization for the expressive power of looped models. First, we show that looped Transformers can effectively solve group composition (a generalization of the addition problem). Then we show a very general result on how a non-looped

model with very few distinct layers can be simulated by a looped model with a small blowup in model size. This result is then used to solve the $p$-hop problem using a one-layer looped transformer. Our construction for group composition and $p$-hop problem are nearly optimal in terms of depth and much more efficient in terms of parameters compared to non-looped models.

## 5.1 Preliminaries and Notations

We first define the standard transformer architecture. Throughout the paper we will fix the dimension of the embedding to be $d \in \mathbb{N}^+$, the vocabulary to be $\mathcal{V}$ and maximum sequence length to be $n_{\max}$. We will use id to denote the identity mapping. Here, we describe the high-level notation. Please refer to Appendix C.1 for detailed notations. We use FF and MHA to denote the feed-forward and attention layers respectively, and $\theta_{\mathsf{MHA}}$ and $\theta_{\mathsf{FF}}$ denote the parameters in these layers.

**Definition 5.1** (Transformer Block). Given number of layers $L \in \mathbb{N}^+$ and parameter $\theta_{\mathsf{TB}} = (\theta_{\mathsf{MHA}}^{(l)}, \theta_{\mathsf{FF}}^{(l)})_{l=0}^{L-1}$, $L$-layer transformer block $\mathsf{TB}_{\theta_{\mathsf{TB}}} : (\mathbb{R}^d)^n \to (\mathbb{R}^d)^n$ for any $n \in \mathbb{N}^+$ is defined as

$$\mathsf{TB}_{\theta_{\mathsf{TB}}} \triangleq (\mathsf{id} + \mathsf{FF}_{\theta_{\mathsf{FF}}^{(L-1)}}) \circ (\mathsf{id} + \mathsf{MHA}_{\theta_{\mathsf{MHA}}^{(L-1)}}) \circ \cdots (\mathsf{id} + \mathsf{FF}_{\theta_{\mathsf{FF}}^{(0)}}) \circ (\mathsf{id} + \mathsf{MHA}_{\theta_{\mathsf{MHA}}^{(0)}}), \quad (5)$$

We also denote EMBED and OUTPUT to be the input embedding and output softmax layers respectively. Please refer to Appendix C.1 for precise definitions. Finally, we define the entire transformer model that maps a sequence of tokens to a distribution over tokens: $p_\theta : \cup_{n \le n_{\max}} \mathcal{V}^n \to \Delta^{|\mathcal{V}|-1}$.

$$p_\theta \triangleq \mathsf{OUTPUT}_{\theta_{\mathsf{OUTPUT}}} \circ \mathsf{TB}_{\theta_{\mathsf{TB}}} \circ \mathsf{EMBED}_{\theta_{\mathsf{TE}}, \theta_{\mathsf{PE}}} \quad (6)$$

where $\theta = (\theta_{\mathsf{TB}}, \theta_{\mathsf{TE}}, \theta_{\mathsf{PE}}, \theta_{\mathsf{OUTPUT}})$ denote all the transformer parameter. In particular, we use $\mathsf{TF}_\theta(v_1, \ldots, v_n) \triangleq \arg\max_{v \in \mathcal{V}} p_\theta(v | v_1, \ldots, v_n)$ to denote the deterministic version of the transformer model. We now define a looped Transformer model that also subsumes a non-looped model.

**Definition 5.2** ($(L \otimes T)$ Looped Transformer). Given the number of loops $T \in \mathbb{N}^+$, parameters $\theta = (\theta_{\mathsf{TB}}, \theta_{\mathsf{TE}}, \theta_{\mathsf{PE}}, \theta_{\mathsf{OUTPUT}})$, where $\theta_{\mathsf{TF}} = (\theta_{\mathsf{MHA}}^{(l)}, \theta_{\mathsf{FF}}^{(l)})_{l=0}^{L-1}$, we define a $(L \otimes T)$ *looped Transformer* as $p_{\theta,T} \triangleq \mathsf{OUTPUT}_{\theta_{\mathsf{OUTPUT}}} \circ (\mathsf{TB}_{\theta_{\mathsf{TB}}})^T \circ \mathsf{EMBED}_{\theta_{\mathsf{TE}}, \theta_{\mathsf{PE}}}$.

## 5.2 Group composition problem

We consider the problem of composing $n$ elements from a group, and prove that a standard 1-layer transformer looped $\mathcal{O}(\log(n))$ times can solve this problem. This is a generalization of the modular addition problem and has a long history (see Appendix C.2.2). Recently, Liu et al. (2022) show that transformers with $\log_2 n$ depth can compute composition over $n$ group elements, regardless of whether the group is solvable or not. However, the construction in Liu et al. (2022) uses different attention parameter for each layer. Here, we provide a more parameter efficient construction where we solve this problem by looping a one-layer transformer $\log(n)$ times. This result matches the depth upper bound shown for non-looped models by Liu et al. (2022), and is very close to their super constant lower bound. Thus looped models can solve the problem with best known depth.

**Theorem 5.1.** For any finite group $G$ and every $n \in \mathbb{N}^+$, there exists a constant-precision looped transformer $\mathsf{TF}_{\theta,T}$ computing the composition of $n$ elements from $G$ with a 1-layer transformer block, $T = \lceil \log_2 n \rceil$ loops, $G \cup \{\#\}$ being the vocabulary, $d = 3(\lceil \log_2 |G| \rceil + \lceil \log_2 n + 1 \rceil)$ embedding size, $d_{\mathsf{FF}} = |G|^2 + 6\lceil \log_2 |G| \rceil$ hidden dimension in MLP, $d_{\mathsf{ATTN}} = \lceil \log_2 n \rceil$ hidden attention dimension, and 2 attention heads. More specifically, for any $g_1, \ldots, g_n \in G$, $\mathsf{TF}_\theta(\#, g_1, \ldots, g_n) = g_1 \circ \cdots g_n$.

## 5.3 Looped models can simulate non-looped models

Our second theoretical result shows that a non-looped transformer with repeated layers can be simulated by a looped transformer with fewer parameters and same depth.

**Theorem 5.2.** For any transformer $p_\theta$ with $L$ layers, $d$ embedding size, $d_{\mathsf{FF}}$ hidden dimension for MLP, $H$ attention heads with $d_{\mathsf{ATTN}}$ hidden dimension, at most $R$ distinct transformer layers and

bounded activation value, there is a looped transformer $p_{\theta',L}$ working on the same vocabulary $\mathcal{V}$ plus a dummy token $\#$, which loops a 1-layer transformer block for $L$ times, with $d + R + 2$ embedding size, $Rh_{\mathsf{FF}} + O(L)$ hidden dimension for MLP, $RH$ attention heads with $d_{\mathsf{ATTN}}$ hidden dimension, such that for any string $v_1, \ldots, v_n \in \mathcal{V}$, $p_\theta(v_1, \ldots, v_n) = p_{\theta',L}(\#, v_1, \ldots, v_n)$.

We can also use the above theorem to convert the $O(\log_2 n)$ depth transformer that simulates group composition into a 1-layer looped transformer, although it is less parameter efficient than the result from the previous section. Please refer to Appendix C.2.1 for the full proof.

**Theory for $p$-hop.** The experiments on $p$-hop induction from Section 2.1 surprisingly show that a small model looped multiple times can solve it very effectively. We establish a theoretical basis for this finding. More specifically, we show that a constant layer transformer with $\log(p)$ loops suffices to solve $p$-hop induction problem. This result, in fact, matches the lower bound for layers required for non-looped models proved in Sanford et al. (2024b). The result follows from Theorem 5.2 and the construction for non-looped models from Sanford et al. (2024b) (restated in Theorem C.12).

**Corollary 5.3.** $p$-hop problem (Definition B.1) can be solved by looping a 1-layer transformer $\lfloor \log_2 p \rfloor + 2$ times, which has $O(\log n)$ bits of precision, $d = d_{\mathsf{FF}} = d_{\mathsf{ATTN}} = O(1)$ embedding size, hidden dimension for MLP and attention, and at most 3 attention heads.

## 5.4 Looped models can simulate CoT reasoning

In Section 3.3, we discussed how looped models can be viewed as generating multiple latent thoughts in each iteration. The following theorem shows that one can use looped transformer with $L$ loops to simulate $L$ steps CoT of another transformer with similar sizes.

**Theorem 5.4** (Looped transformer simulates CoT). For any $L$-layer non-looped transformer $\mathsf{TF}_\theta$ with fixed input length $n$ and the number of CoT steps $m$, there exists a looped transformer $\mathsf{TF}_{\theta'}$ with $L + \mathcal{O}(1)$ layers, $\Omega(\log(n + m))$, more embedding dimension and constantly many more attention heads, such that for any input $\boldsymbol{v} = (v_i)_{i=1}^n$, the output of non-looped transformer after $m$ steps of CoT, $\mathsf{TF}_\theta^m(\boldsymbol{v})$, is the same as that of the looped transformer on input $x$ concatenated by $m$ dummy tokens with $m$ loops, $\mathsf{TF}_{\theta',m}(\boldsymbol{v}, \#^m)$.

We sketch the high-level idea behind Theorem 5.4; the full proof can be found in Appendix C.3.

- (Masking all-but-one) We maintain a counter $t$ (Lemma C.10) at each position for the current number of loop and for the $i$th position, we will only "update" the embedding if $i - n \geq t$ and reset the embedding to the input to the looped transformer layer otherwise. This can be implemented by MLP. So similar to CoT, the first $n + i - 1$ embedding won't be changed in the $i$th loop.
- (Shift by one) We can use attention to obtain the output of the current loop at the previous position and use that as the input of the next loop at current position. (Lemma C.9)
- (Token decoding) We show that we can use MLP with ReLU activation to simulate the encoding and decoding process of CoT. (Lemma C.7)

## 6 Conclusions, limitations and future work

This work explores a new direction of "looped models for reasoning". Not only are looped models able to solve many reasoning problems with very fewer parameters, they also have an inductive bias towards disproportionately improving the reasoning performance of language models, over memorization. The theoretical results on the expressivity of looped models start to provide some hints into their depth-optimality. While we test looped models on a subset of reasoning problems, a natural question is whether the results hold for many other forms of reasoning (e.g. multimodal and common-sense reasoning). In particular, a succinct formalization of reasoning problems itself is an interesting future direction. Furthermore, the inductive bias towards improved reasoning performance at the same perplexity is very intriguing and deserves further exploration. We find the scaling behavior of looped models very fascinating, and the connections to *latent thoughts* and CoT reasoning start to provide hints into this behavior. We hope this inspires future exploration on using looped models for more efficient inference-time scaling to aid with deeper reasoning.

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

## A  RELATED WORK

Reasoning is recognized as a core ability for intelligent and robustly model and has thus seen significant focus over the last few years. The synthetic reasoning problems we consider in this work have all been used in the prior works of Sanford et al. (2024b); Ye et al. (2024); Sanford et al. (2024a); Nogueira et al. (2021) to theoretically analyze the strengths and limitations of Transformers. There is also interest in the representation power of Transformers for computational

problems (Liu et al., 2022; Strobl et al., 2023) and for chain-of-thought reasoning (Merrill & Sabharwal, 2023a; Feng et al., 2023; Li et al., 2024). The necessity of model depth for performance has been remarked upon, for small models (Liu et al., 2024) and reasoning (Chen & Zou, 2024; Ye et al., 2024; Petty et al., 2023). In this work, we make a finer observation that albeit larger depth is necessary, this can be achieved with a limited parameter budget via looping.

Looping in transformer models has been studied since the works (Dehghani et al., 2018; Lan et al., 2020) where they showed the benefits of looping for supervised learning tasks and BERT pretraining respectively. Looping also appears in (Schwarzschild et al., 2021; Bansal et al., 2022) that study the extrapolation properties of looping for certain algorithmic tasks. More recently Giannou et al. (2023); de Luca & Fountoulakis (2024) have studied the theoretical properties of looped decoder models and show that looping can simulate arbitrary Turing machines. In addition Yang et al. (2023); Gao et al. (2024); Gatmiry et al. (2024a;b) study looped models as a way to simulate iterative algorithms for in-context learning. Recently, Mohtashami et al. (2023) introduced CoTFormer, which tries to improve the perplexity of looped language models and (Liu et al., 2024) explore latency-efficiency parameter sharing for on-device LLMs. In contrast, our work focuses on the surprising inductive bias of looping to improve downstream reasoning tasks, and goes beyond algorithmic and in-context learning tasks.

Different training algorithms (e.g. gradient descent (Soudry et al., 2018)) and architectural choices (e.g. attention (Edelman et al., 2022)) have been shown to have certain implicit biases. There is increasing interest in such inductive biases during pretraining (Saunshi et al., 2022; Liu et al., 2023). More recently, Saunshi et al. (2024) showed an inductive bias of stacking (Reddi et al., 2023) towards improving reasoning and hypothesize that a connection of stacking to looped models could be responsible for this. Our results provide further verification for this hypothesis.

## B EXPERIMENTS

### B.1 SIMPLE REASONING SETUP DETAILS

$n$**-ary addition.** All experiments are run on a standard Transformer architecture with input dimension of 256, 8 heads and 1024 hidden dimension in the feed-forward layers. We train using Adafactor (Shazeer & Stern, 2018) employing a linear warmup coupled with a cosine decay schedule for the learning rate. All runs use a batch size of 1024, learning rate of 0.005 and run for 200k steps. This corresponds to 200M examples, which is insignificant compared to the total possible examples ($> 10^{320}$). Thus memorization of the answer is not an issue. Since training is a bit noisy, for each setting, we run 3 different random seeds and pick the run with the maximum average accuracy. We pick maximum instead of average because we care about expressivity power of these models.

$p$**-hop induction.** We formally define the $p$-hop problem below:

**Definition B.1** ($p$-hop, Sanford et al. (2024b)). For a finite alphabet $\Sigma$, define the map $\mathsf{hop}_p : \Sigma^n \to \Sigma \cup \{\perp\}$ as $\mathsf{hop}_p(\boldsymbol{v}) = v_{\mathsf{find}_p(\boldsymbol{v},n)}$ if $\mathsf{find}_p(\boldsymbol{v}, n) \neq 0$ else $\perp$, where

$$\mathsf{find}_1(\boldsymbol{v}, i) = \max\left(\{0\} \cup \{j \leq i, v_{j-1} = v_i\}\right)$$
$$\mathsf{find}_p(\boldsymbol{v}, i) = \mathsf{find}_1(\boldsymbol{v}, \mathsf{find}_{p-1}(\boldsymbol{v}, i)) \text{ for } p \geq 2.$$

For the $p$-hop problem we sample instances randomly while enforcing that there always exists $p$-hops present in the input sequence. We do this by first picking the sequence of $p$-hops randomly and then shuffling them around in a sequence with filler tokens to be filled by the remaining characters. After the shuffle, we sample the remaining characters to occur in place of the filler tokens while respecting the $p$-hop order. Our train set consists of 4M examples and our test and validation sets consist of 262k examples each. For all models we train on this dataset, the model dimension is 128, hidden dimension is 512 and 8 attention heads are used. Rotary positional encodings are used as well. We train using Adafactor for 200,000 steps with a batch size of 256 using a base learning rate of $10^{-3}$ and use a linear warmup coupled with a cosine decay schedule for the learning rate.

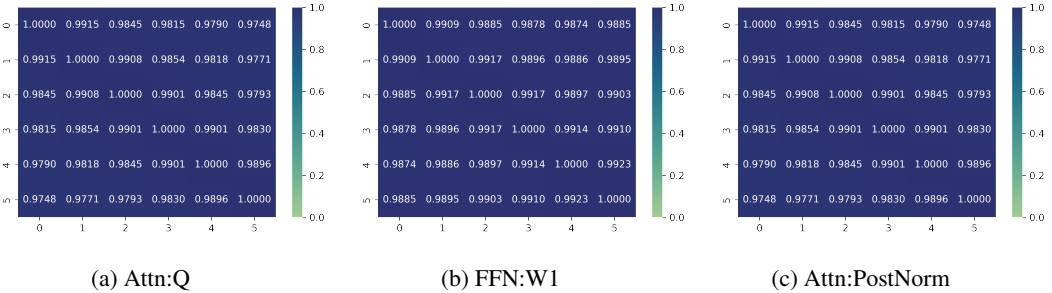

(a) Attn:Q      (b) FFN:W1      (c) Attn:PostNorm

Figure 3: Cosine similarities for different layers in the model trained with the regularization strength $\lambda_{\text{reg}} = 10$ for block size $k = 4$ (see Section 4 for details). The 24 layer model will have 6 such blocks of size 4. The heatmap above shows the cosine similarities between weights for all pairs of blocks. Overall we find the final cosine similarity to be very high, thus suggesting a strong connection to looped models.

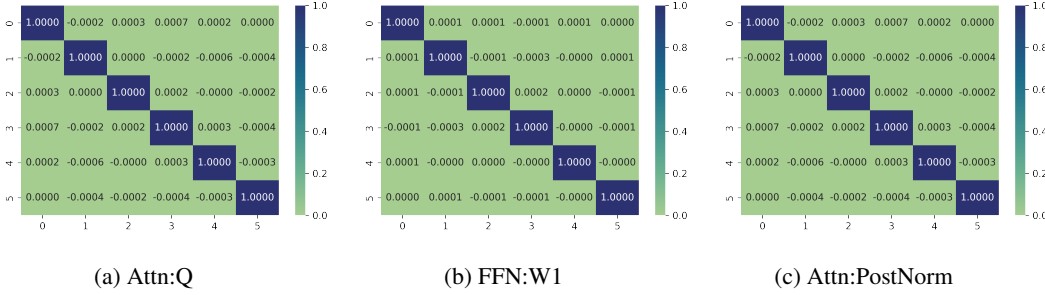

(a) Attn:Q      (b) FFN:W1      (c) Attn:PostNorm

Figure 4: Cosine similarities for different layers in the baseline model trained without any regularization strength. Overall the cosine similarities are very low for large matrices, as expected for high dimensions matrices.

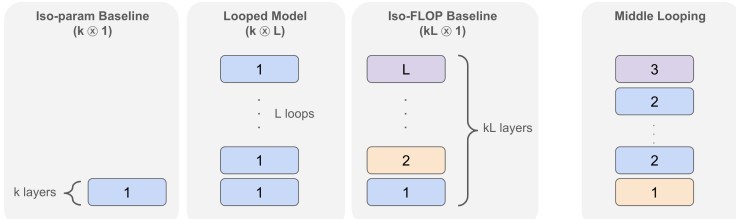

Figure 5: Illustration of the simple and architecture agnostic looping mechanism that we consider. A $k$-layer block looped $L$ times (**middle**) is denoted by $(k \otimes L)$, which can essentially be viewed as a weighted shared model. The iso-param baseline, $(k \otimes 1)$, is a $k$-layer model with the same number of *distinct* parameters. The iso-FLOP baseline, $(kL \otimes 1)$, is a $kL$-layer model with the same depth but $L$ times more parameters. Middle looping is a strategy that is inspired from prior works on model stacking (e.g. (Saunshi et al., 2024)).

**i-GSM.** We describe how the i-GSM dataset is generated in more detail here. We start with a hierarchy of entities of depth 4 from which we build a randomly sampled structure graph where directed edges connect entities in level $i$ to those in level $i + 1$. Each edge in the structure graph defines a instance parameter which is an integer (for e.g. an edge between city center and car parks denotes the number of car parks in city center). We then construct a randomly sampled mathematical dependency graph which is a DAG over all the instance parameters by relating each to the others. Finally we pick one of the nodes of the dependency graph to query and the goal is to compute the numerical value of this node modulo some prime number $P$. For more details on the sampling process for the structure and dependency graphs, we refer the reader to Ye et al. (2024). We make 3 simplifications compared to Ye et al. (2024): we phrase our problems in a symbolic language without the English construction of sentences (see Table 8); we do not allow abstract parameters in our problems; we perform arithmetic modulo 7 as opposed to 23. Our train dataset consists of around 4 million examples and we test on around 50k examples. Given the significantly larger number of unique solution templates possible, train-test overlap in the problem template space is going to be limited with high probability. For all models we train on this dataset, the model dimension is 128, hidden dimension is 512 and 8 attention heads are used. Rotary positional encodings are used as well. We train using Adafactor for 200,000 steps with a batch size of 256 using a base learning rate of $10^{-3}$ and use a linear warmup coupled with a cosine decay schedule for the learning rate.

## B.2 LANGUAGE MODELING SETUP

We train on the Pile data using causal language modeling objective. The dataset is pre-processed and cached to ensure that all models are trained on exactly the same tokens in the same order. For all experiments, we use a batch size of 512 and sequence length of 1280. We use a cosine learning rate schedule decaying over 400k steps with a peak learning rate of 0.01, tuned based on the baseline model. The base model is a 1.5B parameter decoder only Transformer model, with 24 layers, model dimensions of 2048, hidden dimension 5120 and 32 heads. For the shallower baselines and looped models, we only change the number of layers and keep all other hyperparameters the same.

## B.3 RESULTS FOR EACH TASK GROUP

In Section 3 we discussed results for various task groups like closed book QA, math word problems etc. Here we present details on the task groups and results for each individual task for completeness.

- **Closed book QA**: This includes tasks like TriviaQA (Joshi et al., 2017), TydiQA-NoContext (Clark et al., 2020), Natural Questions (Kwiatkowski et al., 2019) and Web Questions (Talmor & Berant, 2018) that test the model's ability to answer questions without any context, and thus, primarily measure the memorization abilities of language models.
- **Open book QA**: This includes tasks like TydiQA-GoldP (Clark et al., 2020), SquadV2 (Rajpurkar et al., 2018), Drop (Dua et al., 2019), QuAC (Choi et al., 2018), CoQA (Reddy et al.,

Table 4: Downstream evaluations for the models in Table 2 for closed book QA tasks.

| | Params / FLOPs | TriviaQA | TydiQA-NoContext | Natural Questions | Web Questions | Average Average |
|---|---|---|---|---|---|---|
| Baseline | 24x / 24x | 24.5 | 10.6 | 4.3 | 5.6 | 11.2 |
| Base $(12 \otimes 1)$ | 12x / 12x | 15.9 | 9.3 | 2.4 | 5.2 | 8.2 |
| Loop $(12 \otimes 2)$ | 12x / 24x | 18.6 | 9.6 | 3.4 | 5.8 | 9.3 |
| Regularized $(k = 12)$ | 24x / 24x | 20.9 | 10.9 | 3.6 | 5.0 | 10.1 |
| Base $(8 \otimes 1)$ | 8x / 8x | 12.3 | 7.4 | 1.8 | 3.9 | 6.3 |
| Loop $(8 \otimes 3)$ | 8x / 24x | 17.3 | 8.8 | 2.5 | 5.2 | 8.5 |
| Regularized $(k = 8)$ | 24x / 24x | 23.6 | 10.6 | 4.1 | 6.7 | 11.3 |
| Base $(6 \otimes 1)$ | 6x / 6x | 7.4 | 4.8 | 1.1 | 2.7 | 4.0 |
| Loop $(6 \otimes 4)$ | 6x / 24x | 16.0 | 9.3 | 2.7 | 4.9 | 8.2 |
| Regularized $(k = 6)$ | 24x / 24x | 25.8 | 12.2 | 4.5 | 5.6 | 12.0 |
| Base $(4 \otimes 1)$ | 4x / 4x | 3.3 | 1.9 | 0.6 | 1.5 | 1.8 |
| Loop $(4 \otimes 6)$ | 4x / 24x | 11.8 | 8.8 | 2.4 | 3.5 | 6.7 |
| Regularized $(k = 4)$ | 24x / 24x | 26.1 | 13.3 | 4.5 | 6.2 | 12.5 |

Table 5: Downstream evaluations for the models in Table 2 for open book QA tasks.

| | Params / FLOPs | TydiQA-NoContext | SquadV2 | DROP | QuAC | CoQA | Average |
|---|---|---|---|---|---|---|---|
| Baseline | 24x / 24x | 33.4 | 41.3 | 23.4 | 18.6 | 52.7 | 33.9 |
| Base $(12 \otimes 1)$ | 12x / 12x | 21.8 | 34.6 | 20.0 | 17.3 | 41.1 | 26.9 |
| Loop $(12 \otimes 2)$ | 12x / 24x | 27.7 | 39.5 | 22.3 | 17.6 | 46.8 | 30.8 |
| Regularized $(k = 12)$ | 24x / 24x | 33.0 | 44.3 | 23.8 | 17.7 | 51.9 | 34.1 |
| Base $(8 \otimes 1)$ | 8x / 8x | 12.7 | 33.8 | 16.0 | 15.3 | 35.9 | 22.7 |
| Loop $(8 \otimes 3)$ | 8x / 24x | 29.8 | 38.1 | 21.6 | 17.6 | 46.6 | 30.8 |
| Regularized $(k = 8)$ | 24x / 24x | 33.0 | 41.7 | 25.2 | 19.3 | 52.9 | 34.4 |
| Base $(6 \otimes 1)$ | 6x / 6x | 8.2 | 26.8 | 14.8 | 14.4 | 32.5 | 19.3 |
| Loop $(6 \otimes 4)$ | 6x / 24x | 26.6 | 34.6 | 20.8 | 18.1 | 43.7 | 28.7 |
| Regularized $(k = 6)$ | 24x / 24x | 34.5 | 46.1 | 26.2 | 18.6 | 53.4 | 35.8 |
| Base $(4 \otimes 1)$ | 4x / 4x | 3.4 | 19.0 | 11.1 | 13.1 | 22.3 | 13.8 |
| Loop $(4 \otimes 6)$ | 4x / 24x | 22.0 | 32.4 | 20.1 | 15.8 | 40.7 | 26.2 |
| Regularized $(k = 4)$ | 24x / 24x | 33.6 | 49.4 | 24.1 | 19.8 | 54.0 | 36.2 |

2019) that evaluate the model's ability to infer the answer to a question from the extra context that is provided, akin to reading comprehension.

- **Math word problems**: To evaluate the model's ability to reason, we test them on math word problems considered in (Wei et al., 2022b). This includes tasks like SVAMP (Patel et al., 2021), ASDiv (Miao et al., 2020), the MAWPS benchmark (Koncel-Kedziorski et al., 2016). We report 5-shot evaluation for the pretrained model on these tasks.
- **Reasoning primitives**: Saunshi et al. (2024) introduced these datasets to study the inductive bias of stacking towards improving reasoning, by isolating simple reasoning abilities. One such primitive is depth-$k$ variable assignment that requires the model to resolve a chain of assignments of length $k$. An example of depth-0 var-assign is *a=1, b=2, c=6, b=?*, and example for depth-1 var-assign is *a=1, b=2, c=a, d=b, d=?*. We evaluate on the math and coding variants of the depth-0 and depth-1 problems using 5-shot evaluation.

Detailed results for closed book QA are in Table 4, open book QA in Table 5, math word problems in Table 6 and reasoning primitives in Table 7. These tables include, both, looped models from Table 2 and the models trained with regularization from Table 3.

### B.4 MIDDLE LOOPING VARIANT AND RELATIONSHIP WITH GRADUAL STACKING

Recently Saunshi et al. (2024) introduced a gradual stacking (Gong et al., 2019; Reddi et al., 2023) approach for training language models called MidAS. This approach gradually grows the model depth as training proceeds by duplicating certain layers of the model in each stacking operation. Surprisingly, they found that MidAS not only speeds up pretraining, but also improves reasoning in the same sense as Figure 1 – better reasoning at the same perplexity. Furthermore, the paper established a strong connection between stacking via MidAS and looped models, owing to the layer duplication operation, and conjectured that this is the reason for such an inductive bias. Our results from the previous section provides a compelling evidence for this conjecture by showing that looped models also show a very similar inductive bias, thus, further strengthening the connection

Table 6: Downstream evaluations for the models in Table 2 for math word problems.

| | Params / FLOPs | ASDiv | MAWPS-AddSub | MAWPS-MultiArith | MAWPS-SingleEq | MAWPS-SingleOp | SVAMP | Average |
|---|---|---|---|---|---|---|---|---|
| Baseline | 24x / 24x | 26.9 | 49.9 | 2.7 | 38.6 | 40.2 | 17.9 | 29.3 |
| Base $(12 \otimes 1)$ | 12x / 12x | 23.1 | 44.8 | 2.0 | 36.8 | 37.9 | 15.3 | 26.7 |
| Loop $(12 \otimes 2)$ | 12x / 24x | 31.5 | 55.9 | 2.0 | 47.4 | 50.2 | 18.5 | 34.3 |
| Regularized $(k = 12)$ | 24x / 24x | 27.4 | 54.4 | 2.7 | 43.7 | 45.9 | 19.8 | 32.3 |
| Base $(8 \otimes 1)$ | 8x / 8x | 12.9 | 32.7 | 2.0 | 19.9 | 21.9 | 13.4 | 17.1 |
| Loop $(8 \otimes 3)$ | 8x / 24x | 23.2 | 54.9 | 2.3 | 36.2 | 38.3 | 15.6 | 28.4 |
| Regularized $(k = 8)$ | 24x / 24x | 31.0 | 56.7 | 3.5 | 40.7 | 47.5 | 17.3 | 32.8 |
| Base $(6 \otimes 1)$ | 6x / 6x | 15.4 | 31.1 | 1.3 | 21.3 | 26.3 | 10.9 | 17.7 |
| Loop $(6 \otimes 4)$ | 6x / 24x | 24.5 | 51.6 | 0.7 | 38.0 | 47.7 | 16.3 | 29.8 |
| Regularized $(k = 6)$ | 24x / 24x | 32.0 | 47.1 | 3.5 | 44.9 | 42.0 | 16.8 | 31.0 |
| Base $(4 \otimes 1)$ | 4x / 4x | 7.9 | 10.4 | 1.5 | 11.0 | 19.0 | 8.3 | 9.7 |
| Loop $(4 \otimes 6)$ | 4x / 24x | 19.9 | 49.1 | 1.7 | 29.3 | 35.9 | 12.6 | 24.8 |
| Regularized $(k = 4)$ | 24x / 24x | 35.0 | 53.9 | 3.2 | 52.2 | 50.9 | 23.0 | 36.4 |

Table 7: Downstream evaluations for the models in Table 2 for reasoning primitives.

| | Params / FLOPs | Code Depth-0 | Math Depth-0 | Code Depth-1 | Math Depth-1 | Average |
|---|---|---|---|---|---|---|
| Baseline | 24x / 24x | 71.1 | 72.5 | 24.2 | 22.3 | 47.5 |
| Base $(12 \otimes 1)$ | 12x / 12x | 52.2 | 51.6 | 20.7 | 18.3 | 35.7 |
| Loop $(12 \otimes 2)$ | 12x / 24x | 73.5 | 86.5 | 21.3 | 23.6 | 51.2 |
| Regularized $(k = 12)$ | 24x / 24x | 75.1 | 74.9 | 27.0 | 25.7 | 50.7 |
| Base $(8 \otimes 1)$ | 8x / 8x | 48.7 | 42.0 | 21.4 | 19.8 | 33.0 |
| Loop $(8 \otimes 3)$ | 8x / 24x | 88.4 | 86.9 | 23.1 | 22.9 | 55.3 |
| Regularized $(k = 8)$ | 24x / 24x | 92.2 | 86.7 | 23.1 | 23.3 | 56.3 |
| Base $(6 \otimes 1)$ | 6x / 6x | 26.3 | 29.7 | 20.2 | 20.1 | 24.1 |
| Loop $(6 \otimes 4)$ | 6x / 24x | 90.6 | 88.1 | 24.1 | 21.7 | 56.1 |
| Regularized $(k = 6)$ | 24x / 24x | 84.0 | 79.7 | 31.4 | 28.3 | 55.8 |
| Base $(4 \otimes 1)$ | 4x / 4x | 19.3 | 23.3 | 17.3 | 17.6 | 19.4 |
| Loop $(4 \otimes 6)$ | 4x / 24x | 87.9 | 90.0 | 24.8 | 24.8 | 56.9 |
| Regularized $(k = 4)$ | 24x / 24x | 88.0 | 86.9 | 27.9 | 25.8 | 57.2 |

between stacking and looped models. Why such an inductive bias occurs is still an open question, and we believe that understanding this is an important future direction.

Furthermore, inspired by their findings, we explore **middle looping** (see Figure 5 for an illustration) — a variant of looping which maintains independent layers at the start and the end of the network, and perform looping on the middle block of layers. The high-level intuition from Saunshi et al. (2024) is that the first and last layers play a special role in the model and thus, should be treated differently from the middle layers. In Table 2, we report results for a version of middle looping that is iso-param with a $(12 \otimes 1)$ baseline and iso-flop with a $(24 \otimes 1)$ baseline, just like the $(12 \otimes 2)$ model. Overall, we find that middle looping has better perplexity and more uniform improvements than the default looping of $(12 \otimes 2)$ (except for math word problems), and thus, might be a more practical looping approach. We leave the exploration of the best looping strategies for future work.

## C    THEORETICAL RESULTS

### C.1    DETAILED NOTATIONS

**Definition C.1** (Embedding Layer). Given a finite vocabulary $\mathcal{V}$, embedding dimension $d \in \mathbb{N}^+$, token embedding parameter $\theta_{\mathsf{TE}} \in \mathbb{R}^{d \times |\mathcal{V}|}$ and position embedding parameter $\theta_{\mathsf{PE}} \in \mathbb{R}^{d \times n_{\max}}$, we define the *embedding layer* as a sequence-to-sequence map, denoted by $\mathsf{EMBED}_{\theta_{\mathsf{TE}}, \theta_{\mathsf{PE}}} : \mathcal{V}^n \to (\mathbb{R}^d)^n$ for any $1 \leq n \leq n_{\max}$, where

$$\mathsf{EMBED}_{\theta_{\mathsf{TE}}, \theta_{\mathsf{PE}}}(v_1, \ldots, v_n) = (\theta_{\mathsf{TE}}(v_1) + \theta_{\mathsf{PE}}(1), \ldots, \theta_{\mathsf{TE}}(v_n) + \theta_{\mathsf{PE}}(n)) . \quad (7)$$

**Multi-Head Self-Attention Mechanism:** Given attention parameters $\theta_{\mathsf{ATTN}} = \{W_Q, W_K, W_V, W_O\}$, where each $W_Q^m, W_K^m, W_V^m, W_O^m \in \mathbb{R}^{d_{\mathsf{ATTN}} \times d}$, we define the *Self-Attention* layer with a causal mask for a decoder-only transformer in Algorithm 1. We also

Table 8: **Left.** Symbolic i-GSM problem and its solution. **Right.** Accuracy of looped and non-looped models on the i-GSM task from Section 2. $(k \otimes L)$ looped model is significantly better than the iso-param $(k \otimes 1)$ model and performs as well as non-looped iso-flop $(kL \otimes 1)$ model.

**Question.** *E#I := 4. E#J := E#I. K#N := I#N + J#O + F#K. F#K := E#J. J#O := F#K + K#O + E#J. H#J := E#J + F#K. I#P := L#M + I#N + K#O. I#M := J#O + J#P + F#K. J#P := H#J - F#K. L#M := I#N + J#P + F#K. I#N := 2 \* J#P + H#J + E#I. K#O := J#P + I#N + E#J. I#P?*

**Answer with CoT.** *E#I = 4.* $\implies$ *E#I = 4. E#J = E#I.* $\implies$ *E#J = 4. F#K = E#J.* $\implies$ *F#K = 4. H#J = E#J+F#K.* $\implies$ *H#J = 1. J#P = H#J-F#K.* $\implies$ *J#P = 4. I#N = 2J#P+2H#J+2E#I.* $\implies$ *I#N = 4. L#M = I#N+J#P+F#K.* $\implies$ *L#M = 5. K#O = J#P+I#N+E#J.* $\implies$ *K#O = 5. I#P = L#M+I#N+K#O.* $\implies$ *I#P = 0.*

| | Params / FLOPs | Accuracy |
|---|---|---|
| Base $(8 \otimes 1)$ | 8x / 8x | **73.2** |
| 1 layer model | | |
| Base $(1 \otimes 1)$ | 1x / 1x | 24.5 |
| Loop $(1 \otimes 2)$ | 1x / 2x | 52.3 |
| Loop $(1 \otimes 4)$ | 1x / 4x | 69.9 |
| Loop $(1 \otimes 8)$ | 1x / 8x | **73.2** |
| 2 layer model | | |
| Base $(2 \otimes 1)$ | 2x / 2x | 54.0 |
| Loop $(2 \otimes 2)$ | 2x / 4x | 66.9 |
| Loop $(2 \otimes 4)$ | 2x / 8x | **73.6** |
| 4 layer model | | |
| Base $(4 \otimes 1)$ | 4x / 4x | 71.3 |
| Loop $(4 \otimes 2)$ | 4x / 8x | **71.6** |

define a *Multi-Head Attention* layer as a collection of self-attention layer with non-shared parameters $\theta_{\mathsf{MHA}} = \{\theta_{\mathsf{ATTN}}^{(h)}\}_{h=1}^{H}$, and its output is the sum of the outputs from each head. That is, $\mathsf{MHA}_{\theta_{\mathsf{MHA}}} = \sum_{h=1}^{H} \mathsf{ATTN}_{\theta_{\mathsf{ATTN}}^{(h)}}.$[2]

---

**Algorithm 1** Causal Self-Attention, ATTN

---

**Input:** Parameter $\theta_{\mathsf{ATTN}} = (W_Q, W_K, W_V, W_O)$, Input embedding $x_1, \ldots, x_n \in (\mathbb{R}^d)^n$.
**Output:** Output embedding $x' = (x'_1, \ldots, x'_n) \triangleq \mathsf{ATTN}_{\theta_{\mathsf{ATTN}}}(x_1, \ldots, x_n)$.
1: $q_i \triangleq W_Q x_i, k_i \triangleq W_K x_i, v_i \triangleq W_V x_i, \forall i \in [n]$
2: $s_i \triangleq \mathrm{softmax}(\langle q_i, k_1 \rangle, \ldots, \langle q_i, k_i \rangle) \| (0, \ldots, 0).$
3: $h'_i \triangleq W_O^\top \sum_{j=1}^{n} (s_i)_j v_j.$

---

**Feed-Forward Network:** Given the parameters of the fully-connected feedforward network layer $\theta_{\mathsf{FF}} = (W_1, b_1, W_2, b_2) \in \mathbb{R}^{x_{\mathsf{FF}} \times d} \times \mathbb{R}^{d_{\mathsf{FF}}} \times \mathbb{R}^{d \times d_{\mathsf{FF}}} \times \mathbb{R}^{d_{\mathsf{FF}}}$, we define the feedforward layer $\mathsf{FF}_{\theta_{\mathsf{FF}}} : \mathbb{R}^d \to \mathbb{R}^d$ as $\mathsf{FF}_{\theta_{\mathsf{FF}}}(h) \triangleq W_2, \mathrm{relu}(W_1 h + b_1) + b_2$.

**Definition C.2** (Output Layer). Given parameter $\theta_{\mathsf{OUTPUT}} \in \mathbb{R}^{d \times |\mathcal{V}|}$, we denote the output layer as $\mathsf{OUTPUT}_{\theta_{\mathsf{OUTPUT}}} : (\mathbb{R}^d)^n \to \Delta^{|\mathcal{V}|-1}$, where

$$\mathsf{OUTPUT}_{\theta_{\mathsf{OUTPUT}}}(x_1, \ldots, x_n) \triangleq \mathrm{softmax}(x_n^\top \theta_{\mathsf{OUTPUT}}) \tag{8}$$

Finally, we define the entire transformer model $p_\theta : \cup_{n \le n_{\max}} \mathcal{V}^n \to \Delta^{|\mathcal{V}|-1}$ as

$$p_\theta \triangleq \mathsf{OUTPUT}_{\theta_{\mathsf{OUTPUT}}} \circ \mathsf{TB}_{\theta_{\mathsf{TB}}} \circ \mathsf{EMBED}_{\theta_{\mathsf{TE}}, \theta_{\mathsf{PE}}} \tag{9}$$

for any $\theta = (\theta_{\mathsf{TB}}, \theta_{\mathsf{TE}}, \theta_{\mathsf{PE}}, \theta_{\mathsf{OUTPUT}})$. For convenience, we also write $[p_\theta(v_1, \ldots, v_n)](v)$ as $p_\theta(v \mid v_1, \ldots, v_n)$. In particular, we use $\mathsf{TF}_\theta(v_1, \ldots, v_n) \triangleq \arg\max_{v \in \mathcal{V}} p_\theta(v | v_1, \ldots, v_n)$ to denote the deterministic version of the transformer model.

**Finite-precision Modeling:** In this paper we assume the transformer is of finite precision. More specifically, we follow the setting in Li et al. (2024) and use the shorthand $\mathbb{F}_s \triangleq \{c \cdot k \cdot 2^{-s} \mid c \in \{-1, 1\}, 0 \le k \le 2^{2s} - 1, k \in \mathbb{N}\}$ to denote fixed-point numbers of constant precision $s$ and rounding operation $[\cdot]_s : \mathbb{R} \to \mathbb{F}_s$ to denote the correcting rounding, namely the mapping from $\mathbb{R}$ to the closest representable number in $\mathbb{F}_s$. (We break the tie by picking the number with smaller absolute value). We assume that (1). all the parameters of the transformer are in $\mathbb{F}_s$ and (2). correct rounding is performed after every binary operation in the forward pass of the transformer. We will refer the readers to Li et al. (2024) for detailed discussion on such finite-precision modeling and only list important notations and lemmas that will be used in this paper below.

---

[2]Though in this paper we focus on attention with casual mask, our definition of looped transformer generalizes to the cases with other attention masks.

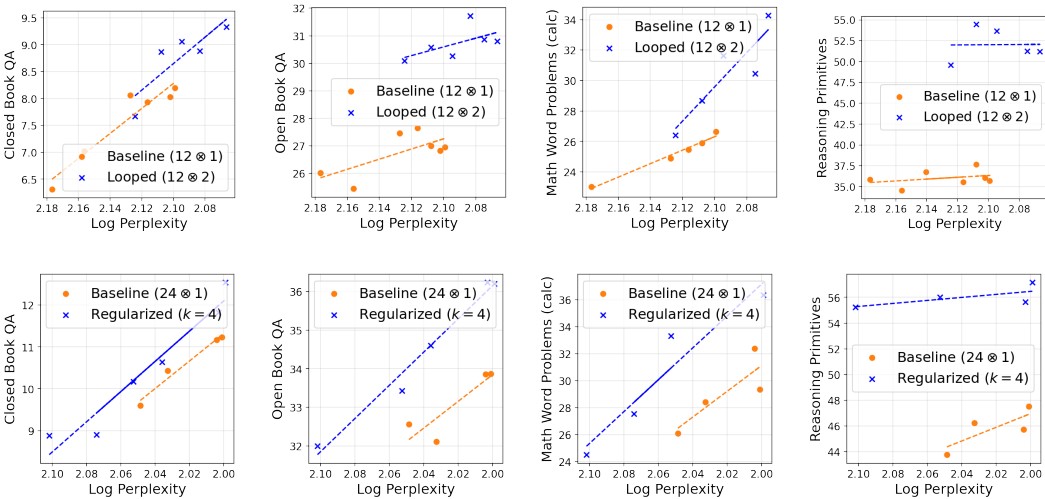

Figure 6: Downstream evaluation for various task groups on the x-axis, vs validation log perplexity on the y-axis (reversed), as training proceeds. The top plots compare a 12-layer baseline model $(12 \otimes 1)$ and the looped model $(12 \otimes 2)$. The second row compares the baseline 24-layer model and the 24-layer model trained with regularization using block size $k = 4$ and $\lambda_{\mathrm{reg}} = 10$ (See Equation (4)). For both comparisons we have similar observations. For closed book QA (memorization) tasks looping has very similar trends to baseline. For open book QA tasks and math word problems, looping has much better downstream performance at an equivalent log perplexity. This verifies the inductive bias of looping and regularization towards better reasoning abilities.

We use $1_s$ to denote all-one vectors of length $s$. Similarly we define $\langle \cdot, \cdot \rangle_s$, $\times_s$, and $\mathrm{softmax}_s$. We recall that for any $s \in \mathbb{N}^+$ and integer $0 \leq x \leq 2^s - 1$, we use $\mathrm{bin}_s(x) \in \{0, 1\}^s$ to denote the usual binary encoding of integer $x$ using $s$ binary bits in the sense that $x = \sum_{i=1}^{s} 2^i (\mathrm{bin}_s(x))_i$ and $\mathrm{sbin}_s(x) \in \{-1, 1\}^s$ to denote the signed binary encoding, which is $2\mathrm{bin}_s(x) - (1, \ldots, 1)$. Finally we define $B_s = \max \mathbb{F}_s = 2^s - 2^{-s}$.

**Lemma C.1.** [Lemma E.1, (Li et al., 2024)] For any $s \in \mathbb{N}^+$, it holds that $\left[ \exp(-B_s) \right]_s = 0$.

**Lemma C.2.** [Lemma E.2, (Li et al., 2024)] For any $s \in \mathbb{N}^+$, it holds that $\left[ \exp(B_s) \right]_s = B_s$.

**Lemma C.3.** [Lemma E.5, (Li et al., 2024)] Unlimited-fanin $\mathsf{AND}, \mathsf{OR}$ (resp. $\mathsf{MAJORITY}$) : $\{0, 1\}^n \to \{0, 1\}$ can be simulated by some 2-layer feedforward ReLU network with constant (resp. $\log n$) bits of precision constant hidden dimension and additional $n$ constant inputs of value 1.

Mathematically, let $\mathsf{FF}[s(n)]$ be the set of functions $C : \{0, 1\}^n \to \{0, 1\}$ which can be a two-layer feedforward ReLU network with at most $s(n)$ bits of precision and constant hidden dimension $\mathsf{FF}_\theta : \{0, 1\}^{2n} \to \{0, 1\}, \mathsf{FF}_\theta(x') = W_2 \times_s \mathrm{relu}([W_1 \times_s x' + b_1]_s)$, where $\theta = (W_2, W_1, b_1)$, such that for any $x \in \{0, 1\}^n$,

$$\mathsf{FF}_\theta(x_1, 1, x_2, 1, \ldots, x_n, 1) = C(x). \qquad (10)$$

We have unlimited-fanin $\mathsf{AND}, \mathsf{OR} \in \mathsf{FF}[1]$ and $\mathsf{MAJORITY} \in \mathsf{FF}[\log n]$.

Given two vectors $x, y$ of the same length $s$, we use $x ^\frown y$ to denote their interleaving, that is, $(x ^\frown y)_{2i-1} = x_i, (x ^\frown y)_{2i} = y_i$ for all $i \in [e]$.

**Lemma C.4.** [Lemma E.3, (Li et al., 2024)] For any $s \in \mathbb{N}^+$, let $q_i = \mathrm{sbin}_s(i) ^\frown 1_s$ and $k_i = B_s \cdot (\mathrm{sbin}_s(i) ^\frown (-1_s))$ for all $i \in [2^s - 1]$, it holds that $\left[ \exp(\langle q_i, k_j \rangle_s) \right]_s = \mathbf{1}[i = j]$ for all $i, j \in [2^s - 1]$.

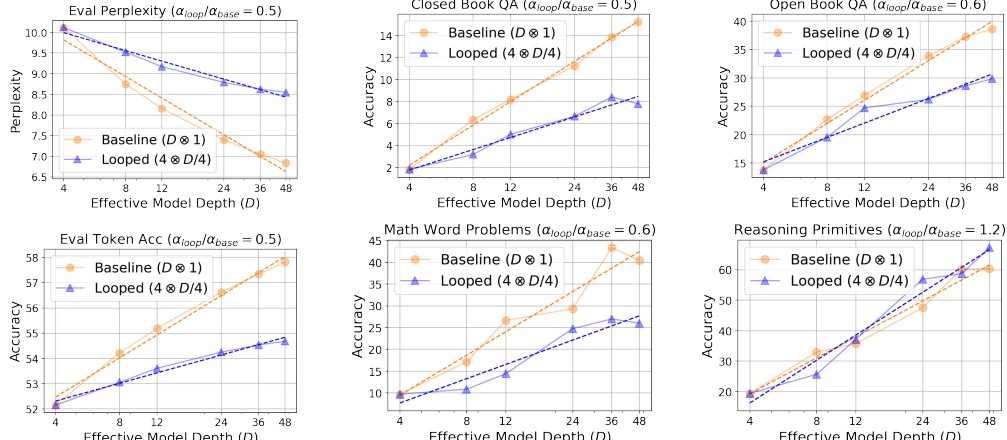

Figure 7: Scaling behavior for various task group as the effective depth increases. The blue curve shows how performance scales as the number of loops increases, without increasing parameters, using models of the form $(4 \otimes D/4)$ for various values of $D$. The orange curve visualizes the scaling behavior of $(D \otimes 1)$ which increases the depth by adding fresh parameters. For reasoning primitives, the looped model scales as well, or even better, than the baseline despite having $D/4$ fewer parameters.

## C.2 PROOFS

### C.2.1 LOOPED MODELS CAN SIMULATE NON-LOOPED MODELS

*Proof of Theorem 5.2.* We start by introduce some more notations. We will proof the theorem for any fixed sequence of $\boldsymbol{v} = (v_1, \ldots, v_n)$. We use $\boldsymbol{x}^{(l)} = (x_i^{(l)})_{i=1}^n$ to denote the intermediate embedding of $p_\theta$ in the $l$th layer. More specifically, we define

$$\boldsymbol{x}^l = (\mathsf{id} + \mathsf{FF}_{\theta_{\mathsf{FF}}^{(l-1)}}) \circ (\mathsf{id} + \mathsf{MHA}_{\theta_{\mathsf{MHA}}^{(l-1)}}) \circ \cdots (\mathsf{id} + \mathsf{FF}_{\theta_{\mathsf{FF}}^{(0)}}) \circ (\mathsf{id} + \mathsf{MHA}_{\theta_{\mathsf{MHA}}^{(0)}}) \circ \mathsf{EMBED}_{\theta_{\mathsf{EMBED}}}(\boldsymbol{v}).$$
(11)

We also use $\boldsymbol{x}^{(l+0.5)} = (x_i^{(l+0.5)})_{i=1}^n$ to denote the intermediate embedding of $p_\theta$ in the $l$th layer after the attention layer.

$$\boldsymbol{x}^{l+0.5} = (\mathsf{id} + \mathsf{MHA}_{\theta_{\mathsf{MHA}}^{(l-1)}})(\boldsymbol{x}^l).$$
(12)

Similarly, for the constructed looped transformer $p_{\theta,T}$, we use $\boldsymbol{v}' = (\#, \boldsymbol{v}_1, \ldots, \boldsymbol{v}_n)$ to denote its input. For simplicity, we use the convention that $\#$ is at position 0. The proof still works if $\#$ starts at position 1 because we can just transfer the later tokens by 1 position. We define $\boldsymbol{x}'^{(l)} = (x_0'^{(l)}, x_1'^{(l)}, \ldots, x_n'^{(l)})$ as the intermediate embedding of $p_\theta$ in the $l$th layer and $\boldsymbol{x}'^{(l+0.5)} = (x_0'^{(l+0.5)}, x_1'^{(l+0.5)}, \ldots, x_n'^{(l+0.5)})$ as the intermediate embedding of $p_\theta$ in the $l$th layer.

Below we first state the key properties that our construction will satisfy, which imply the correctness of the theorem and then we state our construction of $p_{\theta',T}$ and show the properties are actually satisfied:

- $x_i'^{(l)} = (x_i^{(l)}, \mathbf{1}_R - e_{r(l)}, l, \mathbf{1}[i = 0])$.
- $x_i'^{(l+0.5)} = (x_i^{(l+0.5)}, \mathbf{1}_R - e_{r(l)}, l, \mathbf{1}[i = 0])$.
- $x_0^{(l)} = x_0^{(l+0.5)} = \mathbf{0}$.[3]

---

[3]Here we abuse the notation for simplicity of presentation. $x_0^{(l)} = x_0^{(l+0.5)}$ are not defined in the original non-looped transformer. The key point here is that they are 0 vectors throughout the forward pass.

---

**Algorithm 2** Group Composition

---

**Input:** Group elements $g_0, g_1, \ldots, g_n \in G$, where $g_0 = e$.
**Output:** $g_0 \circ g_1 \circ \ldots g_n$.
  1: $g_i^{(0)} = g_i, \forall 0 \le i \le n$.
  2: **for** $l = 1 \to \lceil \log_2 n \rceil$ **do**
  3: $\qquad a_i^{(l)} = g_{[2i-n-1]_+}^{(l-1)}, b_i^{(l)} = g_{[2i-n]_+}^{(l-1)} \ \ \forall 0 \le i \le n$.
  4: $\qquad g_i^{(l)} = a_i^{(l)} \circ b_i^{(l)}, \forall 0 \le i \le n$.
  5: **end for**
  6: **return** $g_n^{(\lceil \log_2 n \rceil)}$.

---

To get the last coordinate $l$, which is a depth counter, we just need to add 1 more hidden dimension in MLP.

Next we show we can use two-layer with $L + 2$ MLP to get the mapping from $\ell \mapsto \mathbf{1}_R - e_{r(l)}$. Let $(\theta_{\mathsf{FF}}^{(i)}, \theta_{\mathsf{MHA}}^{(i)})_{i=1}^R$ be the parameters of the $R$ distinct layers in $\theta$. We assume in the $l$th layer, $r(l)$'s parameters are used. This is because $e_{r(l)} = \sum_{i=1}^{L} e_{r(i)} 0.5 * ([l-i+1]_+ - 2[l-i]_+ + [l-i-1]_+)$.

Now we explain how to deactivate the undesired MLP neurons. In other words, our construction of $\theta_{\mathsf{FF}}'$ is essentially concatenation of $\theta_{\mathsf{FF}}^{(i)}$ for $i \in [r]$ in the hidden dimension of $\mathsf{FF}$, with the additional control that $\mathsf{FF}_{\theta_{\mathsf{FF}}'}((x_i^{(l)}, \mathbf{1}_R - e_{r(l)}, l), \mathbf{1}\,[i = 0]) = \sum_{i=1}^{R} \mathsf{FF}_{\theta_{\mathsf{FF}}^{(i)}}(x_i^{(l)}) \mathbf{1}\,[r(l) = i, i \neq 0]$ at $l$th layer. This control can be done by subtracting $\mathbf{1} - e_{r(l)} + \mathbf{1}\,[i = 0]$ by a constant which is larger than the maximum pre-activation in the hidden layer.

Finally we explain how to deactivate the undesired attention. We will only use attention to update the first part of the embedding, which is $x_i^{(l+0.5)}$. A crucial step here is that we set the token embedding of $\#$ as 0 We construct keys and queries as follows:

1. $W'^{(r')}_Q(x_i'^{(l)}) = (W_Q^{(r')} x_i^{(l)}, 1 - \mathbf{1}\,[r' = r(l)]$ for $r' \in [R]$ and $i = 0, \ldots, n$
2. $W'^{(r')}_K(x_i'^{(l)}) = (W_K^{(r')} x_i^{(l)}, -B\mathbf{1}\,[i = 0])$ for $r' \in [R]$ and $i = 0, \ldots, n$, where $B$ is some constant larger than the maximum previous inner product in attention, $\max_{l \in [L], i, j} \left\langle (W_K x_i^{(l)}, (W_Q x_i^{(l)}) \right\rangle$.
3. $W'^{(r')}_O W'^{(r')}_V (x_i'^{(l)}) = (W_O^{(r')} W_V^{(r')} x_i^{(l)}, \mathbf{0}, 0, 0)$.

This construction works because only the 'desired' attention head $r = r(l)$ will be activated and behave as in the non-looped case, because otherwise all position in that attention head will be completely attended to position 0 and returns a zero vector. (We can choose $B$ to be large enough and distribution calculated by the attention score is delta distribution) at position 0, which yields a zero vector as its value. This completes the proof. $\qquad\square$

### C.2.2 GROUP COMPOSITION.

The landmark result in automata theory, Krohn-Rhodes Decompotision Theorem (Krohn & Rhodes, 1965), shows that all semi-automaton with solvable transformation group (which includes composition problem of solvable groups) can be simulated by a cascade of permutation-reset automata, which can be simulated by $\mathsf{TC}^0$ circuits. (Liu et al., 2022) further showed that such automaton with solvable transformation group can be continuously simulated by constant-depth transformers. However, it is also shown (Barrington, 1986) that the composition problem of unsolvable groups are $\mathsf{NC}^1$-complete, for example, the composition of permutation group over 5 elements, $S_5$. Under the common hardness assumption that $\mathsf{NC}^1 \neq \mathsf{TC}^0$), constant depth transformer cannot solve composition of $S_5$ using a single forward pass (Merrill & Sabharwal, 2023b; Liu et al., 2022; Li et al., 2024). But with CoT, very shallow transformers (depth equal to one or two) can simulate the composition problem of any group(Li et al., 2024; Merrill & Sabharwal, 2023a).

*Proof of Theorem 5.1.* We will set the token embedding of $\#$ the same as that of $e$, which is the identity of $G$. In the following proof, we will just treat $\#$ as $e$. We will construct the transformer simulating group composition following the following algorithm Algorithm 2, which gives the high-level idea of the construction. The correctness of Algorithm 2 follows from the associativity of group composition. More concretely, we can verify by induction that $g_0^l \circ g_1^l \circ \ldots g_n^l$ is the same for all $l = 0, \ldots, \lceil \log_2 n \rceil$ and in the final round, i.e., when $l = \lceil \log_2 n \rceil$, $g_i^{(l)} = e$ for all $i < n$.

Below we show how to construct a transformer of the given sizes to simulate the above Algorithm 2. We will embed each $g \in G$ as a different vector $\overline{g} \in \{-1,1\}^{\lceil \log_2 |G| \rceil}$ and each position $0 \le i \le n$ as its binary representation in $\overline{i} \in \{-1,1\}^{\lceil \log_2 n+1 \rceil}$, which is a shorthand for $\mathsf{sbin}_s(i)$ with $s = \lceil \log_2 n + 1 \rceil$. We concatenate them to get $\{x_i^{(0)}\}_{i=0}^n$, that is, $x_i^{(0)} = (\overline{g_i}, \overline{i}, \overline{[2i-n-1]_+}, \overline{[2i-n-1]_+}, 0^{\lceil \log_2 |G| \rceil}, 0^{\lceil \log_2 |G| \rceil})$. For convenience, we will drop the 0's in the end (also in the other proofs of the paper) and write it as $x_i^{(0)} = (\overline{g_i}, \overline{i}, \overline{[2i-n-1]_+}, \overline{[2i-n-1]_+})$. Below we show we can construct 1-layer transformer block with parameter $(\theta_{\mathsf{MHA}}, \theta_{\mathsf{FF}})$ satisfying that

1. $\left[\mathsf{MHA}_{\theta_{\mathsf{MHA}}}\left((\overline{g_i}, \overline{i}, \overline{[2i-n-1]_+}, \overline{[2i-n-1]_+})_{i=0}^n\right)\right]_k = (0^{\lceil \log_2 |G| \rceil + 3\lceil \log_2 n+1 \rceil}, \overline{g_{[2k-n-1]_+}}, \overline{g_{[2k-n]_+}})$
   for all $g_0 = e, g_i \in G \forall i \in [n], k = 0, \ldots, n$;
2. $\mathsf{FF}_{\theta_{\mathsf{FF}}}(\overline{g}, \overline{i}, \overline{j}, \overline{k}, \overline{g'}, \overline{g''}) = (\overline{g' \circ g''} - \overline{g}, 0^{3\lceil \log_2 n+1 \rceil}, -\overline{g'}, -\overline{g''})$, for all $i, j, k = 0, \ldots, n$, $g, g', g'' \in G$.

The first claim is because we can use two attention heads to retrieve $\overline{g_{[2k-n-1]_+}}$ and $\overline{g_{[2k-n]_+}}$ respectively, where both of them use $\overline{k}$ as the key and use $-\overline{[2k-n-1]_+}$ and $-\overline{[2k-n]_+}$ as queries respectively. This is possible because all the required information are already in $x_i$. We further make attention temperature low enough so the probability returned by attention is a one-hot distribution at the position whose key is equal to the negative query after rounding.

Now we turn to the second claim about MLP. We will use $|G|^2$ neurons with ReLU activation and bias to simulate the product of $g'$ and $g''$. We can index each neuron by $(h, h')$ for $h, h' \in G$ and set its incoming weight $[W_1]_{(h,h'),:} = (\overline{h}, \overline{h'})$ and set bias $(b_1)_{(h,h')} = -2\lceil \log_2 |G| \rceil + 1$, which ensures that the activation of neuron $(h, h')$ will only be 1 when $g' = h, g'' = h'$ and be 0 otherwise. Then setting the outgoing weight of neuron $(h, h')$ as $\overline{h \circ h'}$ and the bias in the second layer to be 0 finishes the construction for simulating the group composition. Finally we use the remaining $6\lceil \log_2 |G| \rceil$ to simulate negative identity mapping $x \to -x$ for the remaining $3\lceil \log_2 |G| \rceil$ embedding dimension. This completes the proof. □

## C.3 CONNECTION TO CHAIN-OF-THOUGHT REASONING

In this section, we establish a connection betwee looped models and CoT reasoning. We first define the recursion for CoT reasoning as follows:

$$\mathsf{TF}_\theta^i(v_1, \ldots, v_n) \triangleq \mathsf{TF}_\theta^{i-1}(v_1, \ldots, v_n, \mathsf{TF}_\theta(v_1, \ldots, v_n)),$$

for $i, n \in \mathbb{N}^+$ satisfying $i + n \le n_{\max} - 1$ along with the base case of $\mathsf{TF}_\theta^1(v_1, \ldots, v_n) \triangleq \mathsf{TF}_\theta(v_1, \ldots, v_n)$. For all $0 \le i \le n_{\max} - n - 1$, the output with $i$ steps of CoT is $v_{n+i+1} = \mathsf{TF}_\theta^{i+1}(v_1, \ldots, v_n) = \mathsf{TF}_\theta(v_1, \ldots, v_n, v_{n+1}, \ldots, v_{n+i})$.

We first give the formal statement below.

**Theorem C.5** (Looped transformer simulates CoT). For any $L$-layer non-looped transformer $\mathsf{TF}_\theta$, there exists a looped transformer $\mathsf{TF}_{\theta'}$ with $L + \mathcal{O}(1)$ layers, constantly many more dimensions in embedding, MLP and attention layers and constantly many more attention heads, such that for any input $\boldsymbol{v} = (v_i)_{i=1}^n$ and integer $m$, the output of non-looped transformer after $m$ steps of CoT, $\mathsf{TF}_\theta^m(\boldsymbol{v})$, is the same as that of the looped transformer on input $x$ concatenated by $m$ dummy tokens with $m$ loops, $\mathsf{TF}_{\theta',m}(\boldsymbol{v}, \#^m)$.

Below are some helping lemmas towards showing Theorem C.5 is at least as powerful as CoT.

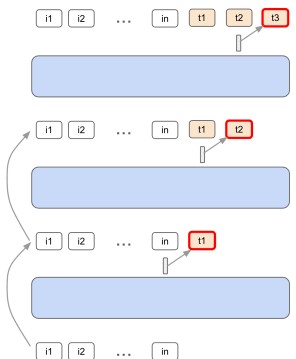 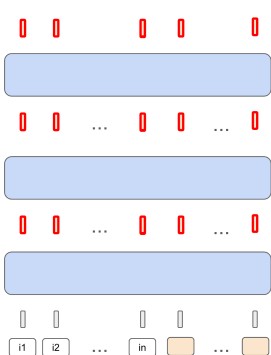

Figure 8: **Left.** Chain-of-thought reasoning can be viewed as a looped model, where each iteration produces one new thoughts token. The new tokens are highlighted in red. **Right.** A looped model can instead generate multiple *latent thoughts* in parallel and, in theory, can simulate CoT reasoning my masking the updates appropriately (see Theorem 5.4)

**Lemma C.6** (Simulating $\arg\max$ using MLP). For every $d \in \mathbb{N}$ and precision $s \in \mathbb{N}^+$, there exists a 3-layer network with relu activation and $d^2$ width $f$ with $s$ precision, such that for any $x \in \mathbb{F}_s^d$, if there is $k \in [d]$, such that $x_k > \max_{j \neq k, j \in [d]} x_j$, $f(x) = e_k$.

*Proof of Lemma C.6.* Define $g_i = 2^s \cdot \mathsf{relu}(2^{-s} - \sum_{j \neq i} \mathsf{relu}(x_j - x_i))$ for each $i \in [n]$. We claim that if there is $k \in [d]$, such that $x_k - \max_{j \neq k, j \in [d]} x_j \geq 2^{-s}$, $g_i = 1$ iff $i = k$ for all $i \in [d]$. First $g_k = 2^s \cdot \mathsf{relu}(2^{-s}) = 1$. Next for $i \neq k$, it clearly holds that $\sum_{j \neq i} \mathsf{relu}(x_j - x_i) \geq 2^{-s}$ and thus $g_i \leq 0$. This construction can clearly be implemented by a 3-layer relu network with $s$ precision. $\square$

**Lemma C.7** (Simulating Decoding and Embedding using MLP). Given any $s$-precision $\theta_{\mathsf{TE}} \in \mathbb{R}^{d \times \Sigma}$ and $\theta_{\mathsf{OUTPUT}}$, there is a 5-layer network $f : \mathbb{R}^d \to \mathbb{R}^d$ with relu activation and $\max(|\Sigma|^2)$ width with $s$-precision, such that for all $s$-precision $x \in \mathbb{R}^d$ which admits unique $\arg\max$ for $v \triangleq \arg\max_{o \in \Sigma}(x^\top \theta_{\mathsf{OUTPUT}})(o)$, it holds that

$$f(x) = \theta_{\mathsf{TE}}(v).$$

*Proof of Lemma C.7.* This is a simple application of Lemma C.6. $\square$

**Lemma C.8** (Control Gate). A 2-layer relu network with precision $s$ can implement $F : \mathbb{F}_s \times \mathbb{F}_s \times \{0, 1\}, F(x, y, M) = Mx + (1 - M)y$.

*Proof of Lemma C.8.* Note that $F(x, y, M) = \mathsf{relu}(x - 2^s \cdot (1 - M)) - \mathsf{relu}(-x - 2^s \cdot (1 - M)) + \mathsf{relu}(y - 2^s \cdot M) - \mathsf{relu}(-y - 2^s \cdot M)$. The proof is completed. $\square$

**Definition C.3** (Transformer Block with Mask). Given number of layers $L \in \mathbb{N}^+$, parameter $\theta_{\mathsf{TB}} = (\theta_{\mathsf{MHA}}^{(l)}, \theta_{\mathsf{FF}}^{(l)})_{l=0}^{L-1}$, and mask function $M : \mathbb{N} \to \{0, 1\}$, we define the $L$-layer *transformer block with mask* $\mathsf{TB}_{\theta_{\mathsf{TB}}, M} : (\mathbb{R}^d)^n \to (\mathbb{R}^d)^n$ for any $n \in \mathbb{N}^+$ as

$$[\mathsf{TB}_{\theta_{\mathsf{TB}}, M}(\boldsymbol{x})]_i \triangleq (1 - M(i))x_i + M(i)[\mathsf{TB}_{\theta_{\mathsf{TB}}}(\boldsymbol{x})]_i \tag{13}$$

**Definition C.4** (Looped Transformer with Mask). Given number of loops $T \in \mathbb{N}^+$, parameters $\theta = (\theta_{\mathsf{TB}}, \theta_{\mathsf{TE}}, \theta_{\mathsf{PE}}, \theta_{\mathsf{OUTPUT}})$, and mask functions $\{M^t\}_{t=1}^T$, where $\theta_{\mathsf{TF}} = (\theta_{\mathsf{MHA}}^{(l)}, \theta_{\mathsf{FF}}^{(l)})_{l=0}^{L-1}$, we define the *looped transformer with mask* as $p_{\theta, T, M} \triangleq \mathsf{OUTPUT}_{\theta_{\mathsf{OUTPUT}}} \circ \mathsf{TB}_{\theta_{\mathsf{TB}}, M^T} \circ \cdots \mathsf{TB}_{\theta_{\mathsf{TB}}, M^1} \circ \mathsf{EMBED}_{\theta_{\mathsf{TE}}, \theta_{\mathsf{PE}}}$ and the corresponding deterministic version as $\mathsf{TF}_{\theta, T, M}(v_1, \ldots, v_n) \triangleq \arg\max_{v \in \mathcal{V}} p_{\theta, T, M}(v | v_1, \ldots, v_n)$.

**Definition C.5** (Shifting Layer). We define the *shifting layer* $\mathsf{SHIFT} : (\mathbb{R}^d)^n \to (\mathbb{R}^d)^n$ as the following for any $d, n \in \mathbb{N}^+$ and $x_1, \ldots, x_n \in \mathbb{R}^d$:

$$\mathsf{SHIFT}(x_1, x_2, x_3, \ldots, x_n) = (x_1, x_1, x_2, x_3, \ldots, x_{n-1}). \tag{14}$$

**Lemma C.9.** For input sequence length up to some integer $n$, $\mathsf{SHIFT}$ could be implemented by a attention layer by concatenating each embedding $x_i$ with $(\mathsf{sbin}_s(i), \mathsf{sbin}_s(f(i)))$, where $n = \lceil \log_2 n + 1 \rceil$.

*Proof of Lemma C.9.* It is equivalent to show we can construct an attention heads which computes $x_{f(i)}$ at each position $i$.

To do so, we just need to invoke Lemma C.6 and use that to set key and query, so position $i$ attends to position $f(i)$. We set value at position $i$ to be $x_i$. This completes the proof. $\quad\square$

**Lemma C.10.** For any positive integer $s > 0$, there is a constant-depth MLP $F$ with $O(s)$ hidden neurons per layer and parameters in $\mathbb{F}_s$, such that for any input $\mathsf{bin}(x) \in \{-1, +1\}^s$, where $0 \le x \le 2^s - 1$, it holds that

$$F(x) = \mathsf{bin}(x + 1).$$

*Proof of Lemma C.10.* By Lemma C.3, it suffices to show that we can simulate $\mathsf{bin}(x) \mapsto \mathsf{bin}(x) + 1$ using $O(s)$ wide, constant-depth boolean circuits with $\mathsf{AND}, \mathsf{OR}, \mathsf{NOT}$ gates with unlimited fan-ins. This is immediate by noting that

$$[\mathsf{bin}(x + 1)]_i = [\mathsf{bin}(x + 1)]_i \oplus \bigwedge_{j=1}^{i-1} [\mathsf{bin}(x + 1)]_j \tag{15}$$

$\quad\square$

**Lemma C.11.** For any positive integer $s > 0$, there is a constant-depth MLP $F$ with $O(s)$ hidden neurons per layer and parameters in $\mathbb{F}_s$, such that for any input $(\mathsf{bin}(x), \mathsf{bin}(y)) \in \{-1, +1\}^s$, where $0 \le x, y \le 2^s - 1$, it holds that

$$F(x, y) = \mathbf{1}\left[x > y\right].$$

*Proof of Lemma C.11.* By Lemma C.3, it suffices to show that we can simulate $\mathsf{bin}(x), \mathsf{bin}(y) \mapsto \mathbf{1}\left[x > y\right]$ using $O(s)$ wide, constant-depth boolean circuits with $\mathsf{AND}, \mathsf{OR}, \mathsf{NOT}$ gates with unlimited fan-ins. This is immediate by noting that

$$\mathbf{1}\left[x > y\right] = \bigvee_{i \in [s]} \left( ([\mathsf{bin}(x)]_i = 1) \wedge ([\mathsf{bin}(y)]_i = 0) \wedge \bigwedge_{1 \le j < i} ([\mathsf{bin}(x)]_j = [\mathsf{bin}(y)]_j) \right). \tag{16}$$

$\quad\square$

*Proof of Theorem C.5.* We consider mask $M^t(i) = \mathbf{1}\left[i - t \ge n\right]$, which we call it CoT masking. Let $s = \lfloor \log(n + m) + 1 \rfloor$, we use $\bar{i}$ to denote $\mathsf{sbin}_s(i)$ for $1 \le i \le n + m$ for convenience. The embedding size of our newly constructed transformer is larger than the target transformer to be simulated by an additive constant of $3s$. We denote the new embedding at position $i$ after $t$th loop by $(x_i^{(t)}, p_i^{(t)})$, where $x_i^{(t)}$ will be the original embedding of the transformer to be simulated, and $p_i^{(t)}$ is of dimension $3s$ and only depends on $i$ and $t$. In particular, we can show that we can set $p_i^{(t)} \triangleq (\bar{i}, \overline{[i - 2]_+ + 1}, \overline{n + t})$ to save information about position and the number of loops — $p_i^{(0)}$ is from the positional embedding and the update is due to Lemma C.10. The size of hidden dimension of MLP and attention (key, query, value) will also be increased by $O(s)$.

The proof contains two steps:

1. To show that there is a transformer with CoT masks simulating the target transformer with $m$ steps of CoT and $L$ layers by looping its own $L + O(1)$ layer block $m$ times.

2. To show that the above looped transformer with CoT masks can be simulated by a standard looped transformer without mask and with constantly many more layers.

For the first claim, starting from the same parameter of the transformers with CoT $\theta$, we build a new looped model with parameter $\theta'$ with constantly many more layers in each transformer block and at most constantly many more heads per attention layer. First, we can add constantly many more layers to use MLP to simulate the decoding-encoding process using Lemma C.7. Next, we can add one more transformer layer in each block and use the attention layer to simulate the shifting layer by Lemma C.9, since we have the additional position embedding $p_i^{(}t)$. In particular, the embedding we get at position $n + t$ after $t$ loops, $x_{n+t}^{(t)}$, now simulates the token embedding of $n + t$ of the CoT transformer. By the way we define CoT mask $M$, for every $t \geq -n + 1$, the embedding $\hat{x}_{n+t}^{(t')}$ will keep the same for all $t' \geq \max(t, 0)$. In $t$th loop, the only embedding update that matters happens at $n + t$th position, because no updates happen at earlier positions, and updates at later positions $n + t'$ for some $t' > t$ will be overwritten eventually in the future loops $t'$, by some value which is independent of their value at the current loop $t$. In the end, we know the embedding $x_i^{(T)}$ in Definition C.4 is exactly equal to that in CoT transformer, and so does the final output.

For the second claim, because CoT mask can be computed by a $O(\log(n + m))$ wide, constant-depth MLP (Lemma C.11), together with Lemma C.8, we know it suffices to increase the number of layers per transformer block and embedding size and hidden dimensions by constant to simulate the transformer with mask by a transformer without mask. $\qquad\square$

**Theorem C.12** (Restatement of Theorem 4.2, (Sanford et al., 2024b)). *$p$-hop problem (Definition B.1) can be solved by $\lfloor \log_2 p \rfloor + 2$-layer non-looped transformer with $\log n$ bits of precision, at most 3 different layers, $d = d_{\mathsf{FF}} = d_{\mathsf{ATTN}} = O(1)$ embedding size, hidden dimension for MLP and attention, 1 attention head.*

