# OpenReview forum: "Reasoning with Latent Thoughts: On the Power of Looped Transformers"
_ICLR.cc/2025/Conference — ICLR 2025 Poster_

### Official Review · Reviewer_iCpz · 2024-10-30

**Soundness:** 3
**Presentation:** 2
**Contribution:** 3
**Rating:** 6
**Confidence:** 3

**Summary:**

The paper investigates the performance of looped Transformers on arithmetic and reasoning tasks and compares it with feedforward models with the same amount of flops. The authors show that looped models can equal and sometimes outperform models with a comparably larger amount of parameters by simply looping over the network, although they demonstrate a worse perplexity. The authors also propose a regularization method to induce feedforward models to behave as looped models and show that it yields better performance on reasoning without reducing perplexity.

**Strengths:**

The experimental setup is sound and the findings are interesting. Although the good performance of looped models is not novel, the paper provide clear insights into the reasons, and makes a disctinction between reasoning and memorization performance. The claims are properly backed-up by the experiments.

The looping-inspired regularization is a particularly interesting and original idea to improve reasoning in Transformers.

The paper is also well-written and easy to follow.

**Weaknesses:**

Experiments lack a comparison with other categories of models that use looping in the computation (e.g. recurrent Transformers [1,2] or Transformers autoregressively returning an output via chain-of-thought or other prompting strategies). Adding comparisons on the arithmetic and reasoning tasks could help to further discriminate what makes looping efficient.

The results in Figure 1 and Table 4 are not thoroughly explained and are hard to interpret. In Figure 1, it is not exactly clear what the y-axis represents and therefore what each data point represents. In Table 4, the choices for the values of $k$ and $\lambda_{\text{reg}}$ are not well justified and a deeper analysis of the results could be helpful.

[1] Bulatov, A., Kuratov, Y., & Burtsev, M. (2022). Recurrent memory transformer. Advances in Neural Information Processing Systems, 35, 11079-11091.

[2] Hutchins, D., Schlag, I., Wu, Y., Dyer, E., & Neyshabur, B. (2022). Block-recurrent transformers. Advances in neural information processing systems, 35, 33248-33261.

**Questions:**

1. In Figure 1, can you explain how each of the data points are generated?

2. In Table 4, do you have an explanation as of why performance increases when setting $k=4$ but then is reduced when $k$ increases?

3. I am curious to know how looping compares with chain-of-thought as the latter effectively performs looping via autoregression but introduces a bottleneck between each loop with the next-token selection. Have you investigated the similarities and differences between the two approaches in your experments?

4. Have you further studied the connection between the number of loops and performance in your experiments, particularly with a high number of loops? Does accuracy keep increasing as the model loops or where is the upper bound? Does it differ depending on the model size?

---

> ### Author Response · Authors · 2024-11-23
> **Author response**
>
> We thank the reviewer for their valuable comments and feedback, and insightful questions. We address them below.
>
> **W1: “Lack a comparison with other categories of models that use looping in the computation (recurrent Transformers, chain-of-thought reasoning)”**
>
> A: We would like to clarify that the motivation and goal of this paper is not to come up with the best kind of looping mechanism. Instead we wanted to show evidence and highlight that **even the simplest of looping mechanisms** can be quite performant in the era of language modeling, if we focus on the right evaluations like reasoning. Having said that, indeed other approaches like recurrent Transformers and CoT reasoning have a similar flavor to looping, and will probably also be effective for reasoning. This is certainly an interesting question, however more detailed evaluation of different methods will require a lot of care and is not the primary motivation of this work. Please also refer to our response to Q3 below for connections with CoT reasoning.
>
> **W2: “The results in Figure 1 and Table 4 are not thoroughly explained and are hard to interpret. …. In Figure 1, can you explain how each of the data points are generated?”**
>
> A: Thank you for the feedback on this. We had included a brief description for Figure 1 in Section 3.2, however we will make this part clearer in the revision. In Figure 1, for each model we evaluate the log perplexity on x-axis and downstream metrics on y-axis. This is evaluated every 20k steps of training, starting from 120k steps, and each point represents one of the checkpoints during pretraining. The downstream metric on the y-axis is an average over accuracies for a task category, e.g. for closed-book QA the y-axis is the average accuracy for TriviaQA TydiQA-NoContext, Natural Questions and Web Questions. We plot these values in a scatter plot as training proceeds. For visualization we plot the best linear fit between log perplexity and the corresponding downstream metric. Plotting this as training proceeds provides insights into how improvements in perplexity translate to improvements in downstream tasks.
>
> In Table 4, we pretrain a 24-layer 1.5B-parameter model with regularization from Section 4. $k$ denotes the block size used for regularization in Eq 4 and $\lambda_{reg}$ is the strength of regularization. $\lambda_{reg}=0$ should recover baseline training.
>
>
> **Q1: “In Table 4, .. why performance increases when setting $k=4$ but then is reduced when $k$ increases?”**
>
> A: This is a good question. While there is no definitive answer for this, our hypothesis is that the inductive bias of looping is stronger where the number of loops is larger. A smaller value of $k$ leads to a higher number of loops ($24/k$). This is most evident from the results on reasoning primitives in Tables 3 and 4 which consistently gets better for smaller values of $k$. Another potential reason could be that hyperparameter choices (learning rate, $\lambda_{reg}$) may not be optimal for the regularization experiments, since they were chosen using the baseline model. It is possible that the optimal learning rate is different for different $k$. We will add this discussion in the paper as well.
>
>
> **Q2: “… the choices for the values of $k$ and $\lambda_{reg}$ are not well justified”**
>
> A: The value of $k$ is picked to be divisors of the total number of layers (i.e. 24). We only tried values larger than 4 to save on compute. We also tried various values of $\lambda_{reg}$ in the range of [0.1, 1, 10, 100]. Smaller values were not leading to any inductive bias, while very large values were understandably causing the training to diverge. In the limit of a very large coefficient, we would end up with fully looped models, which we know would suffer on memorization. Hence we just reported 1 and 10 since these provided a useful signal. Since $\lambda_{reg}=10$ worked best for $k=4$, we picked that for all values of $k$. We will include this discussion in the paper.

---

> > ### Author Response · Authors · 2024-11-23
> >
> > **Q3: “I am curious to know how looping compares with chain-of-thought. Have you investigated the similarities and differences between the two approaches in your experments”**
> >
> > A: Thank you for the question. We indeed touch upon this connection between looped models and chain-of-thought (CoT) reasoning in section B.3. There are some similarities and major differences between the two ideas, as astutely pointed out by the reviewer. One crucial difference, which makes the comparison with looped models unfair, is that for CoT to work we typically require, during training, a different and more extensive dataset containing long reasoning chains rather than just the answer. Looped models, on the other hand, do not require such data and can implicitly learn to reason on their own. Also, we note that looping can technically be complementary to CoT, since a looped model can also be used to predict the full CoT response (as in our experiments with the i-GSM dataset). A deeper and nuanced study of the trade-offs and comparison with CoT requires more careful exploration and is, in fact, a subject of active investigation for us. Any short response here will not do justification to this fascinating question, but we are happy to engage more.
> >
> > **Q4: “Does accuracy keep increasing as the model loops or where is the upper bound? Does it differ depending on the model size?”**
> >
> > A: This is a good question. From what we have seen, the performance does seem to keep improving with more loops. For the simple reasoning tasks in Section 3, even 1 layer model looped $k$ times almost matches a $k$ layer model for the values of $k$ we consider. We are trying to run some more experiments with larger $k$ to understand this better. We will keep you posted on the findings.

---

> > > ### Comment · Reviewer_iCpz · 2024-11-23
> > >
> > > Thank you for your response, my questions have been adequately answered. I will maintain my positive score and look forward to the additional results.

---

> > > > ### Author Response · Authors · 2024-12-02
> > > >
> > > > We thank the reviewer for engaging and are glad that the questions have been adequately addressed. Please find the results of our exploration with #loops below. We note that this exploration and its findings do not impact the main message of the paper, and interestingly, it provides further evidence for the reasoning benefits of looped models. We ran two sets of experiments to test the effect of #loops.
> > > >
> > > > *i-GSM task*: As predicted in our earlier response, the accuracy keeps increasing with more loops and we get almost perfect accuracy after a point. In particular, a 1-layer model looped 16 times gets 96.9% accuracy which is very close to a 16-layer model that gets 97.9%.
> > > >
> > > > *Language modeling*: We additionally tested the effect on #loops by training language models on the PILE dataset. We train a 4-layer model looped $L$ times, i.e. a $(4 \otimes L)$ models, for $L \in \{1, 2, 3, 6, 9, 12\}$. For comparison, we also train non-looped models $(4L \otimes 1)$ models with the same depth $D=4L$, but $L$ times more parameters. In both cases, we find that the accuracies for all task groups continue to increase with more loops/depth, however unsurprisingly the returns are diminishing with depth for both looped and non-looped models. Interestingly, we found that one can fit a simple law that looks like $\text{Acc} = \alpha \log(D) + \beta$ for both looped and non-looped models, where $D$ is the effective depth. $\alpha$ summarizes the impact of depth on downstream performance. For each task group, we measure $\alpha_{loop} / \alpha_{non-loop}$ to see the relative impact of number of loops compared to depth via additional parameters.
> > > >
> > > > | Task category | Eval loss | Closed book QA | Open book QA | Math word problems | Reasoning primitives |
> > > > |---------|----------|----------|-------|---------|----------|
> > > > | $\alpha_{loop} / \alpha_{non-loop}$ | 0.49 | 0.51 | 0.62 | 0.61 | 1.19 |
> > > >
> > > > We find that more loops continue to help, and the relative benefit of loops is higher for tasks that require more reasoning like open book QA and math problems. The impact of loops is even higher (1.19x) than impact of depth for reasoning primitives, which further consolidates the benefit of looped models for reasoning.

---

### Official Review · Reviewer_Ribu · 2024-11-03

**Soundness:** 3
**Presentation:** 3
**Contribution:** 3
**Rating:** 8
**Confidence:** 4

**Summary:**

The work explores the looped models in reasoning tasks, demonstrating that they can achieve deeper reasoning depth with fewer parameters compared to the traditional models. The authors also conducted comprehensive experiments on reasoning tasks like addition and math word problems, and looped models showed improved performance in reasoning, despite having worse performance on memorization tasks. The findings may offer a potential new direction for language model development to improve its reasoning capacity.

**Strengths:**

1. This work provides extensive empirical studies showing that looped models perform well on reasoning tasks, even outperforming traditional models in some cases.
2. This work offers theoretical results that support the expressiveness and efficiency of looped models in solving reasoning problems.
3. The paper is well-presented, making it easy for readers to follow, despite some complex theoretical analysis.

**Weaknesses:**

1. There is limited evidence on the performance of looped models in real-world applications or datasets.
2. It would be much better if the authors could scale up the experiments, and validate the effectiveness of the looped model in real-world tasks.

**Questions:**

please see the weakness.

---

> ### Author Response · Authors · 2024-11-23
> **Author response**
>
> We thank the reviewer for their comments and for valuing the contributions. We address the main concern about real world datasets below by highlighting the extensive evaluations with a 1.5B parameter language model.
>
> **“There is limited evidence on the performance of looped models in real-world applications or datasets…  Authors could scale up the experiments, and validate the effectiveness of the looped model in real-world tasks.”**
>
> A: We would like to direct the reviewer’s attention the results in sections 3 and 4 wherein we trained a 1.5B parameter language model on a real-world language pretraining dataset (Pile), and performed evaluations on 15 real-world benchmark datasets like TriviQA, Squad, and mathematical reasoning datasets like SVAMP and also 4 additional reasoning primitive datasets. Perhaps we misunderstood what the reviewer meant by “real-world applications or datasets” and we are happy to provide a more detailed answer.
> Regarding scaling up experiments further beyond 1.5B, we would like to point out that pre-training experiments are heavily compute hungry (unlike fine-tuning setups which can be more easily performed with 7B sized models) and other leading publications such as “Mamba: Linear Time Sequence Modeling with Selective State Spaces” indeed perform pre-training experiments with similar sized models (up to 2.8B parameters) and similar sized pre-training datasets. We strongly believe that our qualitative insights will extend to larger scale models as well but that would require a significantly larger amount of resources.

---

> ### Comment · Reviewer_Ribu · 2024-11-28
> **Thanks for your response**
>
> Thanks for your response. My concerns have been addressed, I will increase my score.

---

### Official Review · Reviewer_2Phq · 2024-11-04

**Soundness:** 3
**Presentation:** 2
**Contribution:** 3
**Rating:** 6
**Confidence:** 4

**Summary:**

This paper investigates using looped language models for reasoning tasks. The authors demonstrate that reasoning tasks can benefit from deeper models achieved through looping without significantly increasing the parameter count. They show that on synthetic reasoning tasks like multi-digit addition and p-hop induction, looped models can match or even surpass the performance of deeper models with more parameters. Additionally, they identify an inductive bias in looped models toward reasoning tasks and propose a looping-inspired regularization technique to capture these advantages without substantially compromising language modeling efficiency.

**Strengths:**

1. The experiments on synthetic tasks like multi-digit addition and p-hop induction provide a valuable starting point for understanding the strengths of looped models, particularly in recursive reasoning scenarios.

2. Proposing a looping-inspired regularization technique is a noteworthy contribution. It offers a practical method to capture the inductive bias of looped models while maintaining competitive performance in perplexity metrics.

**Weaknesses:**

1. Some explanations, particularly regarding the looping mechanism and training setup, are not as clear as they could be. The paper lacks detailed explanations of how the transformer architecture implements the looping mechanism. For instance, it’s unclear how the KV cache is managed when layers are looped multiple times. Additionally, how does looping interact with the residual connections and layer normalization typically present in transformers? More detailed descriptions and possibly visual aids would enhance understanding.

2. There’s limited discussion on the computational efficiency and scalability of looped models, including potential overhead during training and inference. Specifically, it’s important to know how the training time and inference speed of looped models compare to standard models. Does looping result in longer training times due to repeated computations, or does it impact memory consumption during training and inference?

**Questions:**

1. Given that looped models may have higher perplexity but better reasoning performance, how do you suggest balancing these aspects in practical applications where both are important?

2. Can the looping approach be extended to other domains, such as computer vision models like vision transformers, or models employing different attention mechanisms?

---

> ### Author Response · Authors · 2024-11-23
> **Author response**
>
> We thank the reviewer for their thoughtful comments and for appreciating the various contributions in the paper. Below we address the main concerns about implementation details for looped models. We hope this convinces the reviewers about the setup and its simplicity, and we are happy to engage more.
>
>
> **W1: “The paper lacks detailed explanations of how the transformer architecture implements the looping mechanism”**
>
> A: Thank you for the comment. We will elucidate more details on the looping mechanism in revision. Actually, the looping mechanism is (intentionally) extremely simple and we can simply be viewed as a weight shared model. As described at the start of Section 2, the final model is just a function $f$ composed $L$ times, i.e. $f\circ … \circ f$. Technically $f$ can be any function of choice, with or without residual connection (i.e. $f(x)$ could look like $x + g(x)$) or may or may not have layer norms. The **looping mechanism we consider is oblivious to such architectural choices**. In our experiments we pick $f$ to be a standard $k$-layer Transformer architecture with residual connection and layer norms. As per your suggestion, we will include some visual description of the looping mechanism to make it easier for the reader to understand.
>
> **“.. how the KV cache is managed when layers are looped multiple times”**
>
> A: During decoding, we again treat a looped model as a deeper but weight shared model and use different KV caches for each distinct call to the same layer. So in each loop iteration, for a given layer, we use a fresh KV cache entry so that the previous iterations’ caches are not overwritten.
>
>
>
>
> **W2: “.. limited discussion on the computational efficiency and scalability of looped models, …. Does looping result in longer training times … or does it impact memory consumption during training and inference?”**
>
> A: Training and inference cost for a looped model of effective depth D should be the same as a non-looped model of depth D in terms of FLOPs. In practice, looped models seem to be slightly faster to train, probably because there are fewer parameters to load from memory and also because the backward pass needs to update fewer parameters. This makes looped models especially well suited for extracting higher quality gains in low-memory environments. We will add a discussion of these aspects to the paper. Thank you for your feedback.
>
>
> **Q1: “Given that looped models may have higher perplexity but better reasoning performance, how do you suggest balancing these aspects in practical applications where both are important”**
>
> A: The looping-inspired regularization presented in Section 4 provides a path towards getting better reasoning performance without affecting other evals like perplexity and memorization. A more ambitious approach is to design new architectures that can separate the memory component (that can help with perplexity and memorization) from the reasoning component (through looped models) such that they can work in unison. We believe this is an important next direction but is out of the scope of the current work.
>
> **Q2: “Can the looping approach be extended to other domains? (vision transformer, other attention mechanisms)”**
>
> A: This is a great question and we think this deserves more exploration in future work. Technically the looping mechanism we consider, and its benefits, should be independent of architecture or dataset choices. In this work we restricted all our analyses to text and language models based on Transformer architecture primarily because it is the most common setting used in practice in the context of reasoning.

---

> > ### Comment · Reviewer_2Phq · 2024-11-25
> >
> > Thanks for your detailed response, I will keep my positive ratings.

---

### Official Review · Reviewer_Rw4n · 2024-11-04

**Soundness:** 3
**Presentation:** 3
**Contribution:** 3
**Rating:** 6
**Confidence:** 4

**Summary:**

This paper studies performance of looped transformers on reasoning tasks. They show that looped models result in worse memorization of facts reflected by lower perplexity but can outperform non-looped models on some tasks that require compositional generalization. They highlight that perplexity may be unsuitable metric because looped and non-looped models with same perplexity can result in substantially different performance on downstream tasks requiring reasoning. Lastly, they introduce regularization of non-looped models which forces the weights to be closer to each other in terms of cosine similarity. This leads to a same perplexity but better downstream performance on reasoning tasks.

**Strengths:**

In terms of originality and contribution, the paper demonstrates that a simple regularization can improve reasoning abilities of transformers and provides a basic theoretical analysis of looped models. The writing is clear and the experiments seem sound.

**Weaknesses:**

Personally, I do not find the results very surprising. The fact that looped models can improve performance on algorithmic tasks is known and the extension to the tasks tested in the paper does not seem substantial. It seems to me that that the authors are not aware of the work done in the group of Tom Goldstein related to reasoning abilities of recurrent models (different term for looped models):
https://arxiv.org/abs/2202.05826
https://arxiv.org/abs/2106.04537
And also Daniel Selsam's paper which also showed the importance of recurrence: https://openreview.net/forum?id=HJMC_iA5tm&referrer=%5Bthe%20profile%20of%20Daniel%20Selsam%5D(%2Fprofile%3Fid%3D~Daniel_Selsam1)

Minor:
The first sentence in related work contains a typo: intelligent and robustly model

**Questions:**

1. An obvious goal for the looped models would be to figure out how to make the looping adaptive, i.e. that the model would be able to decide when to stop the looping. Did you considered experiments which would test this ability? This could provide clear benefits over the non-looped baseline and I believe it would be substantial contribution. From Table 3, it is visible that different tasks require different number of loops.
2. Why do the plots in Figure 1 contain datapoints for different perplexities for the two compared models?

---

> ### Author Response · Authors · 2024-11-23
> **Author response**
>
> We thank the reviewer for their useful feedback and appreciation of contributions. The main concerns are addressed below.
>
>
> **“The fact that looped models can improve performance on algorithmic tasks is known and the extension to the tasks tested in the paper does not seem substantial.” “Work done in the group of Tom Goldstein related to reasoning abilities of recurrent models. … Daniel Selsam's paper which also showed the importance of recurrence”**
>
> We thank the reviewer for pointing out the above works. Indeed, they are related to our paper and we will add a discussion on them. In section 5, we also discuss other works on looped models that are in a similar vein. We still believe that our results are novel/surprising and not subsumed by these works. We highlight some key high-level and specific differences below.
>
> - High-level: For algorithmic/reasoning tasks in section 2, the focus is not only on solving them well, but also on the depth-optimality of looped models for those tasks. For instance, a 1-layer model looped 8 times can do as well as an 8-layer model on the i-GSM task. This phenomenon is particularly surprising and novel, and is supported by theory and experiments.
>
> - Moreover the major contribution of this work, that is not covered by any of the prior works, is the role of looped models for language modeling in sections 3 and 4. Most of the focus in the literature has been on perplexity and not so much on downstream task performance. However we uncover an inductive bias that looped models have towards reasoning heavy tasks which we believe is also an interesting and novel finding.
>
> - Comparison to the papers from Tom Goldstein’s group: The two papers by Bansal et al. and Schwarzchild et al. study three synthetic tasks: prefix sum, maze solving and chess puzzles and how well looped ResNet model architectures can generalize to harder problems at test time than those seen at train time on these tasks. While they do demonstrate that looping helps improve the performance of these models on the synthetic reasoning tasks, (a) their main focus is showing the surprising result that looping helps you generalize to out-of-distribution task distributions better, (b) they only consider synthetic examples, (c) they consider non-generative models. In our work we identify an inductive bias of looping towards reasoning tasks in generative language models which are based on the Transformer architecture.
>
> - Comparison to Learning a SAT-Solver from Single-Bit Supervision (Selsam et al.): While this paper resonates deeply with our message on the benefits of looped models, it is a highly stylistic setting in that the architecture is specially tuned for the problem of SAT. And the message passing MPNN used in their paper is quite different from a transformer in how a forward pass works and in particular is not a generative model. Given the generality of the Transformer architecture (not tuned for any specific problem) and the generative nature of our setting we believe our results are novel and different from those in this paper.
>
>
>
> **Q1: “An obvious goal for the looped models would be to figure out how to make the looping adaptive …  Did you consider experiments which would test this ability?”**
>
> A: Adaptivity in looped models is an important topic, and some prior works like Universal Transformers [Dehghani et al. 2018] have touched upon it for algorithmic tasks. For language modeling, our very preliminary experiments do suggest that some level of adaptivity can be achieved. However a more systematic and thorough study would be required for this and it is beyond the scope of this work. We do believe it’s an important future direction to pursue.
>
> **Q2: Why do the plots in Figure 1 contain datapoints for different perplexities for the two compared models?**
>
> A: In Figure 1,  for each model we evaluate the log perplexity on x-axis and downstream metrics on y-axis. This is evaluated every 20k steps of training, starting from 120k steps, and each point represents one of the checkpoints during pretraining. The downstream metric on the y-axis is an average over accuracies for a task category, e.g. for closed-book QA the y-axis is the average accuracy for TriviaQA TydiQA-NoContext, Natural Questions and Web Questions. We plot these values in a scatter plot as training proceeds. For visualization we plot the best linear fit between log perplexity and the corresponding downstream metric. Plotting this as training proceeds provides insights into how improvements in perplexity translate to improvements in downstream tasks.

---

> > ### Comment · Reviewer_Rw4n · 2024-11-26
> >
> > Thank you for your response, my questions have been adequately answered. I'm maintaining my previous rating.

---

### Meta-Review · Area_Chair_BUAU · 2024-12-20

**Metareview:**

The paper explores the use of 'looped models' in reasoning tasks, arguing that often depth, rather than the number of parameters, is crucial for solving many reasoning problems. The authors demonstrate that a k-layer transformer model looped L times can match or outperform a kL-layer non-looped model on various synthetic reasoning tasks such as addition, variable assignment, and math problems. They provide theoretical evidence supporting the effectiveness of looped models and show that these models exhibit an inductive bias towards reasoning tasks, performing well even with fewer parameters. Additionally, they propose a looping-inspired regularization technique that enhances reasoning performance without significantly impacting perplexity.

The main strengths of the paper are:
* A novel application of looped models for reasoning tasks, supported by both theoretical and empirical evidence.
* Comprehensive experiments on synthetic and real-world datasets demonstrate the effectiveness of looped models.
* The paper is well-written and clearly presents the motivation, methodology, and results.

In turn, its main weaknesses are:
* Some findings are known or closely related to existing ones, e.g., some reviewers noted that the benefits of looped models for algorithmic tasks are known, and the extension to the tested tasks may not be substantial.
* The paper lacks detailed explanations of the looping mechanism and its interaction with transformer architecture components like KV cache and residual connections, although this has been partially mitigated during the rebuttal process.
* There is limited discussion on the computational efficiency and scalability of looped models, including potential overhead during training and inference.
* Some missing baselines, especially alternative models that use looped computation.

Overall, the strengths of the paper do outweigh its weaknesses, which explains why all reviewers eventually provided above-acceptance-threshold scores. I agree with their assessment and recommend acceptance for this paper. Furthermore, I believe this paper will be of significant interest to the ICLR community, given how it opens up new avenues for research in adaptive looping mechanisms and the application of looped models to other domains.

**Additional Comments On Reviewer Discussion:**

This paper prompted a moderate amount of discussion between the authors and the reviewers. A summary of their interactions:

#### Reviewer Rw4n
* Points Raised: The reviewer questioned the novelty of the results and suggested exploring adaptive looping mechanisms.
* Author Response: The authors acknowledged related works and highlighted the novel aspects of their study. They also mentioned preliminary experiments on adaptive looping.
* Final Decision: The reviewer’s concerns were adequately addressed, and they maintained a positive rating.

#### Reviewer 2Phq
*Points Raised: The reviewer requested more details on the looping mechanism and its computational efficiency.
*Author Response: The authors provided additional explanations and discussed the efficiency of looped models.
* Final Decision: The reviewer was satisfied with the responses and maintained a positive rating.

#### Reviewer Ribu
* Points Raised: The reviewer suggested validating the effectiveness of looped models on real-world tasks.
* Author Response: The authors pointed to their evaluations on real-world datasets (Sections 3 and 4) and discussed the challenges of scaling up experiments.
* Final Decision: The reviewer increased their score after the authors’ response.

#### Reviewer iCpz
* Points Raised: The reviewer requested comparisons with other models using looping mechanisms and clarification of experimental results.
* Author Response: The authors clarified their experimental setup and discussed the connections with other looping mechanisms.
* Final Decision: The reviewer maintained a positive score after their questions were addressed.

Overall, I believe the authors effectively addressed (most of) the reviewers’ concerns, and the paper’s contributions to the understanding and application of looped models for reasoning tasks justify its acceptance.

---

### Decision · Program_Chairs · 2025-01-22

Accept (Poster)